# ALE-Bench: A Benchmark for Long-Horizon Objective-Driven Algorithm Engineering

**Yuki Imajuku[1], Kohki Horie[1,2], Yoichi Iwata[3], Kensho Aoki[3],**
**Naohiro Takahashi[3], Takuya Akiba[1]**
[1]Sakana AI, Japan    [2]The University of Tokyo, Japan    [3]AtCoder, Japan
{imajuku, takiba}@sakana.ai

## Abstract

How well do AI systems perform in algorithm engineering for hard optimization problems in domains such as package-delivery routing, crew scheduling, factory production planning, and power-grid balancing? We introduce *ALE-Bench*, a new benchmark for evaluating AI systems on score-based algorithmic programming contests. Drawing on real tasks from the AtCoder Heuristic Contests, ALE-Bench presents optimization problems that are computationally hard and admit no known exact solution. Unlike short-duration, pass/fail coding benchmarks, ALE-Bench encourages iterative solution refinement over long time horizons. Our software framework supports interactive agent architectures that leverage test-run feedback and visualizations. Our evaluation of frontier LLMs revealed that while they demonstrate high performance on specific problems, a notable gap remains compared to humans in terms of consistency across problems and long-horizon problem-solving capabilities. This highlights the need for this benchmark to foster future AI advancements.

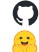 **Code**  github.com/SakanaAI/ALE-Bench
🤗 **Data**  hf.co/datasets/SakanaAI/ALE-Bench

## 1 Introduction

The progress of AI is breathtaking. Benchmarks that were in the spotlight only a few years ago often saturate in performance and quickly lose relevance. To keep advancing AI, we continually need fresh benchmarks that can measure improvements across many facets. In particular, complex end-to-end tasks with long time horizons are expected to form the next frontier of benchmarking [1].

Competitive coding has been one of the most prominent domains for strengthening and evaluating LLMs. Benchmarks such as APPS [2], CodeContests [3], and LiveCodeBench [4] have played a pivotal role. The domain is naturally suited to benchmarking because solutions can be judged automatically and objectively by running code. However, LLM performance in these benchmarks has risen steeply, already rivaling advanced human contestants and showing signs of saturation [5].

Competitive coding falls into two broad categories. The first is the *short-duration, exact-solution contests*, such as the International Olympiad in Informatics (IOI), Codeforces, and the AtCoder Regular Contest (ARC). Here, the problems have intended solutions, and submissions are graded strictly as correct or incorrect. Most previous benchmarks in the AI community have focused exclusively on this category. The second is the **long-duration, score-based contests**, exemplified by the AtCoder Heuristic Contests (AHC) and Topcoder Marathon Matches. Here, the tasks are optimization problems whose true optima are computationally out of reach (e.g., because the underlying problems are NP-hard), and participants spend weeks iteratively refining their programs to push their scores higher. See Figures 1 and 2 and Section 3.1 for details.

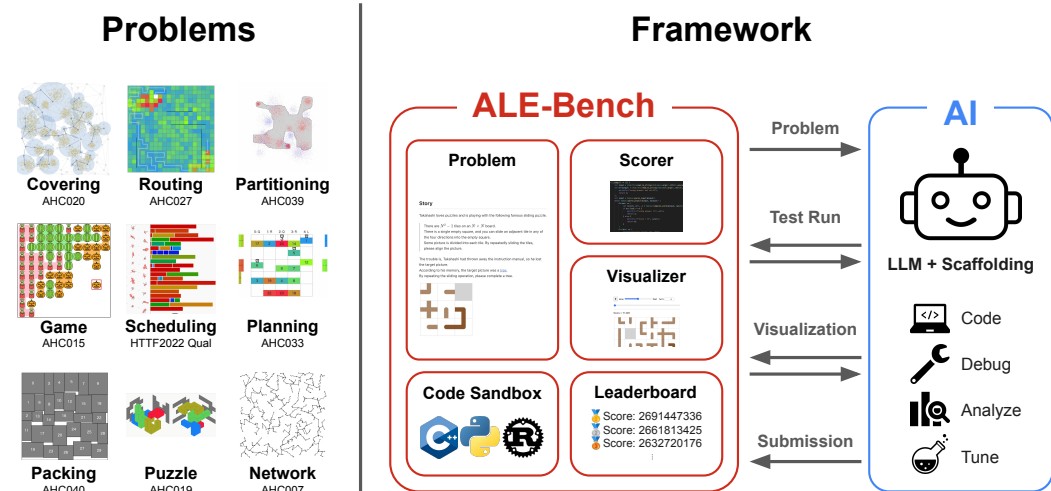

Figure 1: **Overview of *ALE-Bench*. (Left)** ALE-Bench collects past AtCoder Heuristic Contest tasks, hard optimization problems such as routing and scheduling with no known optimum, and ranks submitted programs by score. **(Right)** ALE-Bench covers evaluation from bare LLMs to scaffolded agents. An agent receives a task and submits code. It can optionally invoke test runs and visualization utilities during this process to iteratively refine its solution like a human participant.

This paper introduces **ALE-Bench** (ALgorithm Engineering Benchmark), the first benchmark that measures AI performance in these algorithmic score-based programming contests. We collect past tasks from the AtCoder Heuristic Contest (AHC), one of the world's largest score-based competitions, and provide a software framework for evaluating AI systems on them (Figure 1).

This benchmark quantifies the practical impact of AI systems on algorithm engineering for complex optimization problems within industrial domains, including package-delivery routing, crew scheduling, factory production planning, and power-grid balancing. The tasks in AHC effectively mirror real-world optimization challenges. Each AHC sees thousands of skilled human contestants, including professional experts, invest substantial effort over weeks. This rigorous process allows for performance comparisons against strong human experts. Furthermore, the scoring protocols and evaluation environment in ALE-Bench are standardized to closely replicate those of the actual competitions, thereby enabling direct and equitable comparisons with humans. ALE-Bench is co-developed with AtCoder, which guarantees that all data are fully licensed and that the evaluation procedure faithfully reproduces the original contest conditions.

Beyond these direct applications, the benchmark's second role is to probe the advanced reasoning ability of frontier AI. Just as previous competitive programming benchmarks have served as leading indicators of LLM reasoning capabilities, ALE-Bench, a more challenging competitive programming benchmark, can fulfill a similar function. ALE-Bench is challenging and intriguing because it necessitates longer-horizon reasoning capabilities compared to previous coding benchmarks. Human contestants continuously think and accumulate insights through trial and error over weeks, progressively improving their scores (see Figure 3). A key question is whether AI can demonstrate similar long-horizon reasoning and continuous solution improvement. This benchmark is open-ended, in the sense that true optima are out of reach and scores can keep rising. This allows for meaningful evaluation even after AI systems surpass the performance of the strongest human experts.

We evaluated the performance of frontier LLMs in a one-shot setting and with long iterative refinement using scaffoldings, including ALE-Agent, which we specifically designed for ALE-Bench. Although these models achieved exceptionally high performance on certain problems, the distribution of their contest-wise performance reveals a significant gap compared to true experts' consistency across problem types and of long-term improvement, indicating a need for further AI progress in this field, which ALE-Bench helps to cultivate.

This paper makes two key contributions: ① We propose ALE-Bench for evaluating AI on long-horizon, score-based optimization tasks through interactive, iterative problem solving. ② We analyze current AI's reasoning and algorithm engineering for these complex problems using ALE-Bench.

Figure 2: Example problem from ALE-Bench (`ahc006`). See Figure A1 for the full version.

Write a program that, given a large collection of pickup-delivery pairs on a 2D grid, chooses a prescribed number of requests and outputs a depot-to-depot tour that visits the pickup location of each selected request before its corresponding drop-off. The score is the total length of the route; the shorter, the better.
(CPU time limit: 2 seconds per input)

## 2 Related Work

Early code-generation benchmarks were dominated by unit-test-based suites such as HumanEval [6] and MBPP [7], which contain small, self-contained programming tasks. As LLMs began to exhibit stronger reasoning abilities, researchers shifted toward harder algorithmic problems drawn from competitive programming platforms, e.g., the International Olympiad in Informatics (IOI), Codeforces, and the AtCoder Regular Contests (ARC). This trend gave rise to benchmarks like APPS [2], CodeContests [3], USACO-Bench [8], and LiveCodeBench [4]. On these binary-accuracy benchmarks, frontier-level LLMs already approach the performance of top human contestants. Unlike those benchmarks, ALE-Bench tackles score-based tasks from the AtCoder Heuristic Contest (AHC) that have no single ground-truth answer.

Longer-horizon programming benchmarks that require more than producing a short code snippet include SWE-Bench [9], MLE-Bench [10], and TheAgentCompany [11]. While SWE-Bench and TheAgentCompany remain pass/fail evaluations, both MLE-Bench and ALE-Bench allow for score-based observation of continuous improvement. MLE-Bench is drawn from Kaggle competitions and therefore assesses data-centric machine learning skills, thus operating in a domain different from ALE-Bench. Furthermore, MLE-Bench requires a GPU environment and its evaluation is costly, whereas ALE-Bench uses only a CPU environment and is resource-friendly.

A line of research tackles combinatorial optimization problems with machine learning approaches such as graph neural networks and reinforcement learning [12, 13]. Those studies train specialized models for a single, pre-defined task, where the model receives an instance as input and directly outputs a solution. Our setting is fundamentally different because the tasks are not pre-defined, described in natural language, and solved by writing code. To the best of our knowledge, the only prior work that has studied problems from the score-based contest is FunSearch [14], which explores using LLMs to refine portions of a human-written template. While works like EoH [15] and ReEvo [16] also focus on solving NP-hard problems with LLMs, they do not address problems from score-based contests. Our benchmark is distinct in that it enables a large-scale performance comparison against thousands of human contestants, providing a unique perspective on the capabilities of LLMs.

## 3 ALE-Bench

### 3.1 Dataset Construction

ALE-Bench is built upon the *AtCoder Heuristic Contest (AHC)*, a prominent and one of the largest score-based algorithmic competitions organized by AtCoder Inc. It is held approximately 10 to 18 times annually, with 49 rated contests conducted as of May 1, 2025. Each contest typically attracts around 1,000 participants, and over 6,000 users have been active within the past two years.

A novel problem is introduced at the start of each contest. The problem domains in AHC are diverse, encompassing areas such as routing, planning, multi-agent control, puzzle-solving, and Bayesian inference. As an illustrative example, Figure 2 describes a routing problem asked in the contest. The tasks are typically CPU-bound, with execution time limits ranging from 2 to 10 seconds per case. AHC offers two contest formats: **short** (approximately 4 hours) and **long** (1–2 weeks). The problem characteristics and difficulty significantly differ between these formats. Short contests sometimes involve problems solvable with relatively standard algorithmic approaches like simulated annealing

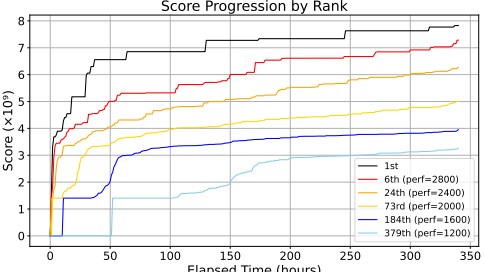

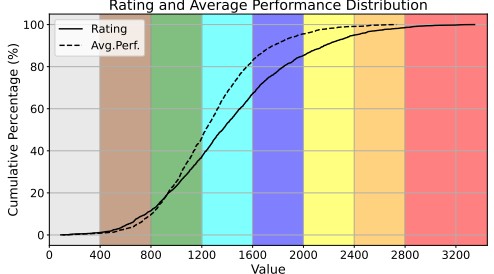

Figure 3: **Long-horizon score ascent in AHC**. Scores at specific ranks at each time point over the two-week AHC014 contest show continual improvement. Line colors mark the color tiers, e.g., perf=2800 (6th) and perf=1200 (379th).

Figure 4: **Rating and average performance distributions.** Cumulative rating and average performance distribution for users with at least 5 participations as of May 1, 2025. Background colors indicate rating tiers.

or beam search. In contrast, long contests often present problems where success hinges on a deeper, iterative process. However, in both formats, achieving high scores requires problem-specific reasoning and refinement through repeated testing. To support this iterative process, visualizers are provided in the contest. As the contest progresses, participants can keep submitting increasingly better solutions. Figure 3 illustrates such score progression during the contest. Moreover, participants often surpass the best in-contest score by continuing to improve their solutions in an official post-contest.

To construct ALE-Bench, we released a dataset of 40 AHC problems on the Hugging Face platform. These problems are sourced from contests conducted up to the end of April 2025. We call the entire set of problems the **full** version, while we also prepared the **lite (subset)** version that contains 10 representative problems. Each problem package contains four elements. (1) *Problem*: a Markdown statement together with any images. (2) *Scorer*: a Rust program evaluating the code on input cases. (3) *Visualizer*: a web-based tool and a Rust program that displays the behavior of the code on the inputs. The image used in Figure 2 is the example visualization. (4) *Leaderboard*: ranking data used for calculating performance metrics. All data are officially provided by AtCoder Inc., ensuring clear licensing and safe use. Additional details are given in Appendix A.1.

## 3.2 Benchmark Implementation

ALE-Bench translates AHC problems into a benchmark tailored for AI systems. We deliver this benchmark as a Python library that leverages the dataset described in Section 3.1. Its core objective is to allow AI systems to fluidly simulate participation in AHC, closely emulating the human contestant experience. To this end, ALE-Bench provides an interface for actions available during a contest, complemented by a code sandbox mirroring the execution environment on AtCoder. This section elaborates on the implementation details. More specific information can be found in Appendix A.2.

Understanding ALE-Bench's design necessitates a brief overview of the typical AHC participant workflow. Typically, an AHC participant selects a programming language from numerous options, develops their solution, and submits it to the online system. This submission is then run in AtCoder's environment, providing instant feedback on a hidden test set of 50 to 300 cases. Final rankings are determined by a **private evaluation** conducted post-contest. The nature of the test cases for this evaluation can vary: a distinct and larger set of hidden inputs for long contests, while the in-contest leaderboard, based on the small test set, serves as the final result for short contests. Additionally, AHC offers participants an input case generator, facilitating local evaluation on either self-generated or provided inputs. This local case evaluation is referred to as **public evaluation**.

ALE-Bench is designed to replicate this AHC environment for AI systems. An AI's evaluation on a single problem is orchestrated by a `Session` object. Its creation triggers a real-time timer, simulating the contest's duration. Within this timed session, as depicted in Figure 1, the AI can undertake several actions via the `Session` : (1) *View Problem*: Access and review the problem statement. (2) *Test Run*: Execute its solution code within the sandbox to get scores from the scorer (a *public evaluation*). The AI can also generate and test new input cases and obtain scores from the scorer for these cases. (3) *Visualization*: Visualize its code's behavior on specific inputs. (4) *Submission*: Submit its final

solution, which initiates the *private evaluation* and the calculation of performance metrics (detailed in Section 3.3). The AI can iterate through actions (1)-(3) as needed within the time limit. The session concludes upon either the submission of a final solution (4) or the expiration of the timer. Post-session, the `Session` object provides the performance metrics from the private evaluation.

A key component is the **code sandbox**, which emulates the AHC execution environment. It currently supports C++, Python, and Rust, which are the most popular choices among AHC participants, and is designed for straightforward extension to other languages. The **visualization** tool further aids development and debugging, offering two modes: (a) Static image generation, optionally producible for each test case during a test run. (b) Interactive web-based visualization, facilitated by a local HTTP server managed by `Session` . This enables more in-depth, interactive visual analysis via a web browser, akin to human contestant practices.

Reproducibility and fair comparison are critical. Given that high-performing AHC solutions are often CPU-intensive and their results can be hardware-dependent, ALE-Bench includes scripts to establish a standardized evaluation environment. These scripts configure Amazon EC2 C6i Instances [17] to mirror the original AtCoder CPU environment. This standardization ensures reproducible results and allows for equitable comparisons among different AI systems and against human performance.

### 3.3  Evaluation Metrics

To comprehensively evaluate the capabilities of AI systems, we employ (i) fine-grained metrics on individual problems, and (ii) aggregated metrics for overall proficiency. These two metrics are primarily derived from the evaluation system used for human contestants in AHC. Our framework facilitates the computation of them, thereby enabling fair and consistent comparisons among AI systems and against human participants. We also provide a detailed description in Appendix A.3.

**Fine-grained Metrics per Problem.**  For each problem, we record (1) the problem-specific score, (2) the *rank*, and (3) the *performance*. Performance is a score derived from a set of participants and the rank within a problem, using an Elo-rating-like method [18]. This score typically ranges from 0 to 3500 and higher is better. Among these three metrics, the performance is particularly useful as it provides a problem-agnostic scale for comparing participants' relative standings.

**Aggregated Metrics across Problems.**  To provide a holistic view of an AI system's abilities over a set of problems, we utilize two primary aggregated metrics: (1) the ***average performance*** and (2) the ***rating*** [19]. The average performance is the simple arithmetic mean of the performances, and its cumulative distribution is shown as the dotted line in Figure 4. On the other hand, the rating is an indicator extensively used on AtCoder to compile leaderboards for human contestants. The rating reflects an individual's overall skill, derived from their performances in a series of contests they have participated in. The line in Figure 4 illustrates the distribution of ratings among active AtCoder users as of May 2025. Since ratings can be significantly lower than a user's actual skill level when the number of participations is very small, we focus on users with at least 5 participations, which is the number officially recommended by AtCoder to ensure rating accuracy.

For evaluating AI systems, *we strongly recommend using average performance as defined above*. The rating design philosophy primarily targets human participants, aiming to: (a) discourage selective submission of solutions only when high ranks are anticipated, and (b) encourage the pursuit of high-risk, high-reward strategies. Despite its utility for human comparable rankings, the rating is less appropriate for evaluating AI systems for two reasons. First, a single exceptionally high performance can disproportionately inflate an AI's rating, potentially leading to an overestimation of its general capabilities. Second, our evaluation protocol involves comparing AI systems with a fixed set of problems. In this context, the rating offers little additional insight beyond the average performance.

In addition to the average performance, examining the performance distribution can also be highly informative. It can reveal whether an AI system's strength lies in excelling at a few specific types of problems or if it demonstrates consistent and broad improvements across the entire benchmark suite.

## 4   ALE-Agent

How much headroom is there for agent-based scaffolding in algorithm engineering? To gain an initial glimpse of the research space opened up by ALE-Bench, we conduct an exploration of special-purpose

agents designed for algorithm engineering. We introduce and develop *ALE-Agent*, a specialized prototype designed as a strong baseline for this new research area. By incorporating established techniques like domain knowledge and inference-time scaling, ALE-Agent moves beyond simple Self-Refine. Its significant performance improvement highlights how such techniques can amplify LLM capabilities, while also demonstrating ALE-Bench's capacity to evaluate these advanced, long-term problem-solving agents.

This algorithm engineering domain has a few distinctive characteristics. For many problem categories, canonical high-level approaches are already known, and choosing the right overall strategy matters enormously. However, even with the right idea, implementation details, hyperparameters, and micro-optimizations can dramatically affect the result. Considering these points, we implement two techniques in the ALE-Agent prototype. See Appendix B for details. These techniques are evaluated in Section 5.3.

**Method 1: Prompting with domain knowledge.** We inject expert knowledge about standard techniques in algorithm engineering, directly into the prompts, such as simulated annealing and beam search. The prompt explicitly discusses search space and evaluation function design, neighborhood generation, and common acceleration tricks.

**Method 2: Diversity-oriented solution search.** We employ a best-first-search-based algorithm to generate and refine answer candidates using an LLM. To avoid prematurely discarding promising solution paths, we augment best-first search with a beam-search-like expansion that spawns multiple children from each node at once. This breadth helps retain high-potential hypotheses and, in practice, amortizes API latency by parallelizing candidate generation, a significant benefit, particularly when working with large reasoning models.

# 5 Experiments

Experiments were conducted on Amazon EC2 C6i Instances [17], mirroring the AtCoder CPU environment. We evaluated up to 22 models from OpenAI [20–25], Google [26–30], Anthropic [31, 32], and DeepSeek [33, 34]. Our primary experimental setup relied solely on text-based feedback from test runs and did not utilize the visualization, except for the OpenHands [35]. Sections 5.1 and 5.2 list the experimental results for the full set, while Section 5.3 lists the results for the lite subset. Further details are in Appendix C.1.

## 5.1 One-Shot Setting

**Setup.** This experiment assessed the one-shot capability of stand-alone LLMs. Before the private evaluation, a public evaluation was performed. If scoring failed due to errors (e.g., compilation, formatting), feedback was provided, allowing up to five code generations. Experiments were conducted in C++20, Python3, and Rust to assess language impact. Further details appear in Appendix C.2.

**Results.** Table 1 shows C++20 results for selected models, and Table 2 compares languages. At the model level, o3-high was the only model to surpass the average performance of 1000. Reasoning models generally outperformed non-reasoning ones. Among non-reasoning models, Claude 3.7 Sonnet performed best. The performance distribution confirmed o3-high's lead, achieving $\geq 400$ performance on all but one problem. However, even top-performing models exceeded 1600 performance on at most 5% of problems and never reached 2000, highlighting the difficulty of ALE-Bench. Ratings did not always align with average performance, e.g., for o4-mini-high vs. Gemini 2.5 Pro. Cost-wise, the reasoning model o4-mini-high was cost-effective, while the non-reasoning Claude 3.7 Sonnet was relatively expensive. In the language comparison, C++20 achieved the highest average performance, rating, and the greatest number of problems with $\geq 400$ performance. However, Rust slightly exceeded C++20 in problems with $\geq 1600$ performance. Python3 and Rust showed similar trends, despite a notable difference in problems with $\geq 1600$ performance. Regarding cost, C++20 was the cheapest per problem but the most expensive per response. This suggests that while C++20 may generate effective solutions more quickly, it does so with higher token consumption per response.

Table 1: **Comparison of frontier LLMs in the one-shot setting.** *Average Perf.* details the average performance on short-/long- format problems and overall problems, respectively. *Perf. Distribution (%)* indicates the percentage of problems for which the performance of $\geq 400$, $\geq 1600$, and $\geq 2000$ were achieved. *Rating* shows the raw value and its corresponding percentile rank from the top. *Cost ($)* lists the incurred USD per problem and per response, respectively. See Table A3 for results of more models and Table A4 for human statistics by levels.

| Model | Average Perf. | | | Perf. Distribution (%) | | | Rating | | Cost ($) | |
|---|---|---|---|---|---|---|---|---|---|---|
| | short | long | overall | $\geq 400$ | $\geq 1600$ | $\geq 2000$ | raw | rank (%) | /problem | /response |
| *Non-Reasoning Models:* | | | | | | | | | | |
| GPT-4o | 547 | 636 | 585 | 75.0 | 0.0 | 0.0 | 936 | 80.1 | 0.048 | 0.022 |
| GPT-4.1 mini | 779 | 755 | 769 | 90.0 | 0.0 | 0.0 | 1135 | 67.4 | 0.009 | 0.005 |
| GPT-4.1 | 696 | 746 | 717 | 80.0 | 2.5 | 0.0 | 1164 | 65.1 | 0.083 | 0.035 |
| Gemini 2.0 Flash | 547 | 585 | 563 | 62.5 | 2.5 | 0.0 | 1031 | 74.6 | **0.006** | **0.002** |
| Claude 3.7 Sonnet | 851 | 810 | 833 | 90.0 | 0.0 | 0.0 | 1197 | 63.2 | 0.287 | 0.142 |
| DeepSeek-V3 | 638 | 688 | 659 | 75.0 | 0.0 | 0.0 | 1142 | 66.8 | 0.008 | 0.003 |
| *Reasoning Models:* | | | | | | | | | | |
| o3-high | **1116** | **946** | **1044** | **97.5** | **5.0** | 0.0 | **1456** | **43.2** | 0.734 | 0.506 |
| o4-mini-high | 866 | 808 | 841 | 92.5 | 0.0 | 0.0 | 1194 | 63.6 | 0.041 | 0.037 |
| Gemini 2.5 Flash | 905 | 827 | 872 | 82.5 | **5.0** | 0.0 | 1422 | 45.5 | 0.194 | 0.104 |
| Gemini 2.5 Pro | 938 | 688 | 832 | 82.5 | **5.0** | 0.0 | 1373 | 49.3 | 0.472 | 0.242 |
| Claude 3.7 Sonnet (Thinking) | 911 | 792 | 860 | 90.0 | 2.5 | 0.0 | 1328 | 52.3 | 0.375 | 0.170 |
| DeepSeek-R1 | 713 | 822 | 760 | 75.0 | 0.0 | 0.0 | 1206 | 62.3 | 0.063 | 0.028 |
| Human Average | 1252 | 1257 | 1260 | 95.7 | 23.8 | 8.5 | 1414 | - | - | - |

Table 2: **Comparison of code languages in the one-shot setting.** For each language, the table reports the mean value over the 22 LLMs used in the one-shot setting experiment.

| Code Language | Average Perf. | | | Perf. Distribution (%) | | | Rating | | Cost ($) | |
|---|---|---|---|---|---|---|---|---|---|---|
| | short | long | overall | $\geq 400$ | $\geq 1600$ | $\geq 2000$ | raw | rank (%) | /problem | /response |
| C++20 | **664** | **672** | **668** | **75.6** | 1.0 | 0.0 | **1072** | **70.7** | **0.144** | 0.086 |
| Python3 | 610 | 643 | 624 | 70.9 | 0.1 | 0.0 | 1024 | 74.1 | 0.161 | **0.075** |
| Rust | 632 | 583 | 611 | 69.4 | **1.1** | 0.0 | 1027 | 73.1 | 0.168 | 0.082 |

## 5.2 Iterative-Refinement Setting

**Setup.** This experiment evaluated the iterative-refinement capability of each LLM. This simple yet widely-adopted method serves as an intermediate step between the basic one-shot setting (Section 5.1) and the advanced agents (Section 5.3). Four models iteratively refined their C++20 solutions for four hours, receiving public evaluation feedback after each attempt, a process similar to Self-Refine [36]. More information is given in Appendix C.3.

**Results.** Table 3 shows that o4-mini-high performed best, achieving an average performance of 1520 and a rating of 2104 (top 11.8% of humans). Performance distributions reveal GPT-4.1 mini and o4-mini-high achieved $\geq 400$ performance in all problems, and all models surpassed 2000 performance on at least one problem. GPT-4.1 mini, the sole non-reasoning model, had fewer $\geq 2000$ performances and a lower average performance than DeepSeek-R1. Average performance improved by over 400 points across all models under the iterative-refinement setting, demonstrating its effectiveness.

**Score Progression.** Figure 5 illustrates o4-mini-high's behavior on `ahc041`, showing its four-hour public score trajectory and generated code file size. Red points mark updates. The score steadily improved, with significant gains even mid-run. The code size also progressively increased, indicating an incremental implementation and refinement process based on feedback, similar to humans.

## 5.3 Scaffolding Evaluation

**Setup.** Effective LLM-based coding relies on both the base model and the scaffolding providing implementation support. This experiment assessed the combined LLM-scaffolding performance. We evaluated OpenHands' CodeActAgent [35] as a representative example of general-purpose scaffolding. Its web browser interaction capability was tested with the ALE-Bench visualization

Table 3: **Comparison of frontier LLMs in the iterative-refinement setting.** The four models listed in the table were tested. *Average Perf.* details the average performance on short-/long- format problems and overall problems, respectively. For the overall average, its corresponding percentile rank from the top is reported. *Perf. Distribution (%)* indicates the percentage of problems for which the performance of $\geq$400, $\geq$1600, $\geq$2000, and $\geq$2400 were achieved. *Rating* shows the raw value and its corresponding percentile rank from the top. *Cost ($)* lists the incurred USD per problem and per response, respectively.

| Model | Average Perf. | | | | Perf. Distribution (%) | | | | Rating | | Cost ($) | |
|---|---|---|---|---|---|---|---|---|---|---|---|---|
| | short | long | overall | rank (%) | $\geq$ 400 | $\geq$ 1600 | $\geq$ 2000 | $\geq$ 2400 | raw | rank (%) | /problem | /response |
| GPT-4.1 mini | 1293 | 1114 | 1217 | 51.5 | **100.0** | 17.5 | 2.5 | 0.0 | 1636 | 30.5 | 2.137 | **0.010** |
| o4-mini-high | **1677** | **1307** | **1520** | **22.3** | **100.0** | **32.5** | **15.0** | **5.0** | **2104** | **11.8** | 7.174 | 0.047 |
| Gemini 2.5 Pro | 1389 | 1301 | 1352 | 36.8 | 95.0 | 27.5 | 7.5 | **5.0** | 1960 | 15.7 | 11.126 | 0.134 |
| DeepSeek-R1 | 1268 | 1155 | 1220 | 51.1 | 97.5 | 15.0 | 5.0 | 2.5 | 1891 | 18.3 | **1.141** | 0.024 |

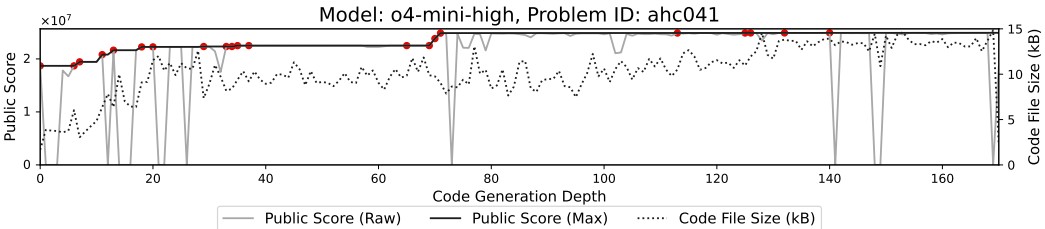

Figure 5: **Trends in public score and code file size in the iterative-refinement setting.** The plot shows the progression of generated code file sizes alongside the corresponding public evaluation scores over a four-hour period. Points farther to the right represent the later time points.

web server. An ablation study of our ALE-Agent (Section 4) used Gemini 2.5 Pro, starting with a base version and incrementally adding Method 1 (domain knowledge) and Method 2 (increased code generation). We conducted only this ablation study with the lite version of ALE-Bench. All runs were limited to 4 hours using C++20. We provide further details in Appendix C.4.

**Results.** Table 4 summarizes results, including iterative-refinement outcomes from Section 5.2. We report the converted lite setting values from the full setting for Sequential Refinement and OpenHands in the table. The full result is shown in Table A5. The average performance of OpenHands showed marginal improvement, which was similar to one-shot results. Cost data suggest OpenHands often exited prematurely, indicating difficulties with improving. In contrast, the base ALE-Agent matched iterative-refinement results. Method 1 slightly improved, while Method 2 yielded substantial gains.

### 5.4 Analysis

**Comparison with Human Experts.** For example, looking solely at ratings in Table 3, o4-mini-high is positioned in the top 11.8% of humans. Does o4-mini-high truly possess abilities comparable to a top-11.8% human expert? Here, we discuss how ratings likely overestimate AI's skills and what areas have potential for future growth.

Its performance distribution significantly differs from humans in this rating band (2050-2150). o4-mini-high achieved $\geq$2400 performance in 5.0% of problems, while humans in this band averaged 4.0%. Conversely, o4-mini-high achieved $\geq$1600 performance in 32.5% of problems; this is below the 56.2% average for humans in the same band and is closer to the 33.9% average for users rated 1600–1700. See Table A4 for a more detailed human performance distribution. This indicates LLMs' significant disparity between strengths and weaknesses. This is further supported by the relatively low 22.3% rank percentile for the average performance. Rating is designed to reward high scores without penalizing low ones (see Section 3.3), thus it tends to overstate AI. To comprehensively assess AI's capability, average performance and performance distribution are more informative.

What causes this gap? Performance trends, for instance, vary between short- and long-format contests. Tables 3 and 4 show AI's short contest performance is typically higher than in long contests. Unlike

Table 4: **Comparison of scaffolding on the lite subset.** Column definitions follow those in Table 3.

| Scaffolding | Average Perf. | | | | Perf. Distribution (%) | | | | Rating | | Cost ($) | |
|---|---|---|---|---|---|---|---|---|---|---|---|---|
| | short | long | overall | rank (%) | ≥ 400 | ≥ 1600 | ≥ 2000 | ≥ 2400 | raw | rank (%) | /problem | /response |
| *Sequential Refinement (Section 5.2) :* | | | | | | | | | | | | |
| GPT-4.1 mini | 1012 | 1021 | 1016 | 73.4 | **100.0** | 0.0 | 0.0 | 0.0 | 990 | 77.4 | 2.12 | 0.008 |
| o4-mini-high | 1449 | 1373 | 1411 | 31.2 | **100.0** | 10.0 | 0.0 | 0.0 | 1386 | 48.2 | 7.22 | 0.047 |
| Gemini 2.5 Pro | 1160 | 1237 | 1198 | 54.1 | **100.0** | 0.0 | 0.0 | 0.0 | 1195 | 63.5 | 11.10 | 0.157 |
| *OpenHands [35]:* | | | | | | | | | | | | |
| GPT-4.1 mini | 600 | 635 | 618 | 96.9 | 80.0 | 0.0 | 0.0 | 0.0 | 650 | 94.2 | **0.15** | **0.004** |
| o4-mini-high | 874 | 845 | 859 | 86.7 | 90.0 | 0.0 | 0.0 | 0.0 | 894 | 83.2 | 2.26 | 0.050 |
| Gemini 2.5 Pro | 726 | 1080 | 903 | 82.8 | 80.0 | 0.0 | 0.0 | 0.0 | 1038 | 74.0 | 3.25 | 0.134 |
| *ALE-Agent w/ Gemini 2.5 Pro (Section 4) :* | | | | | | | | | | | | |
| Base | 1121 | 1213 | 1167 | 56.9 | **100.0** | 10.0 | 0.0 | 0.0 | 1219 | 61.2 | 7.64 | 0.114 |
| + Method 1 | 1448 | 1079 | 1264 | 46.7 | **100.0** | 20.0 | 10.0 | 0.0 | 1494 | 40.2 | 11.12 | 0.135 |
| + Method 1&2 | **2285** | **1474** | **1879** | **6.8** | **100.0** | **70.0** | **30.0** | **20.0** | **2222** | **8.6** | 100.33 | 0.113 |

Table 5: **Comparison between Iterative-Refinement and 150 independent One-Shot trials.** For One-Shot trials, statistics (mean, standard deviation, minimum, quartiles, and maximum) are shown.

| ProblemID | Iterative-Refinement | Mean | SD ($\sigma$) | Min | Q1 | Median | Q3 | Max |
|---|---|---|---|---|---|---|---|---|
| ahc005 | 2107 | 1281 | 216 | 604 | 1138 | 1144 | 1492 | 1722 |
| ahc006 | 2472 | 998 | 308 | 116 | 800 | 1017 | 1259 | 2174 |
| ahc012 | 2236 | 445 | 329 | 123 | 123 | 466 | 697 | 1388 |
| ahc020 | 2545 | 1061 | 104 | 579 | 1015 | 1015 | 1157 | 1731 |
| ahc041 | 2306 | 988 | 251 | 444 | 852 | 1006 | 1074 | 1911 |
| ahc044 | 2150 | 774 | 300 | -80 | 713 | 713 | 713 | 1831 |

humans, who might explore only a few approaches in a short contest, LLMs can rapidly generate and test numerous approaches, potentially outperforming humans through sheer volume. For example, o4-mini-high in Table 3 generated more than 100 solution codes, and ALE-Agent in Table 4 generated approximately 1000. Trying such a large number of solution codes is unrealistic for humans, even in long-format contests. Matching more sophisticated solutions humans devise in long contests remains a greater challenge for AI. Furthermore, analysis of problem types indicates AI excels in contests suited to specific approaches, like simulated annealing. See Appendix C.3 for details.

**Long-Horizon Problem-Solving vs. Massively Parallel Search.** Does achieving a high score through numerous refinements reflect true "long-horizon problem-solving," or simply a "massively parallel search" of mediocre ideas? While our benchmark aims to measure the former, its structure could unintentionally reward the latter.

To investigate this critical distinction, we conducted an additional experiment comparing the performance of the Iterative-Refinement strategy against a series of independent One-Shot trials. We selected six problems where o4-mini-high achieved ≥2000 performance in the Iterative-Refinement setting and executed the One-Shot (C++20) setting 150 times for each problem. The number 150 chosen to match the average number of code generations in the Iterative-Refinement experiments. The results are summarized in Table 5. While the parallel search occasionally reached a respectable score (e.g., 2174 on ahc006), the vast majority of attempts resulted in much lower performance. Notably, the peak performance observed across 150 independent trials was consistently and significantly inferior to the scores achieved through iterative refinement, with the performance deficit being approximately 300 points or more.

This substantial performance gap strongly suggests that simply exploring a large number of independent ideas is insufficient for achieving top-tier results on ALE-Bench. Instead, success demands the ability to refine and improve solutions based on feedback. This thus reinforces the benchmark's capacity to measure genuine long-horizon reasoning skills beyond mere brute-force exploration.

**Contamination.** To investigate potential training data contamination, we analyzed performance variations based on contest dates, as shown in Figure 6. Each model's performances are plotted against its knowledge cutoff date (red line). No model exhibited notable performance changes around its cutoff, suggesting minimal contamination effects.

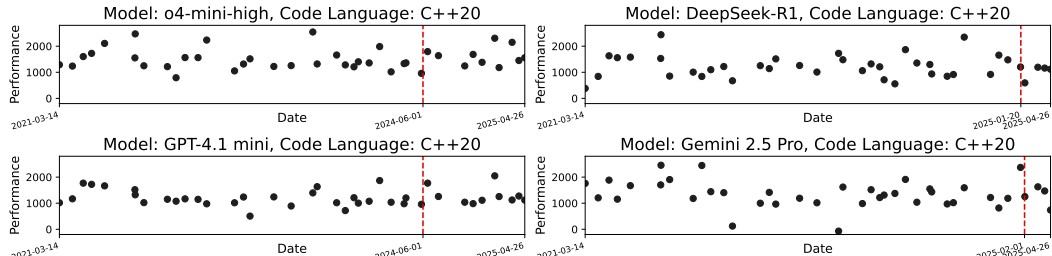

Figure 6: **Investigation of contamination.** For each model, a scatter plot is shown with contest end dates on the x-axis and performance on the y-axis. The red vertical line indicates the knowledge cutoff date of the model. For DeepSeek-R1, its release date was used as official information is unavailable.

**Plagiarism.** To address potential plagiarism concerns, an AHC organizer (one of the authors) manually reviewed all 12 AI-generated programs that achieved $\geq 2000$ performance in the iterative-refinement setting. The organizer, familiar with leading solutions, found no instances where AI-generated code showed stylistic or logical resemblance close enough to suggest plagiarism of human submissions. We thus conclude that performance overestimation due to plagiarism is unlikely.

# 6 Conclusion

We introduced ALE-Bench, a new benchmark that evaluates an AI system's ability to develop and refine algorithms, and we compared both one-shot and iterative-refinement performance across several state-of-the-art LLMs and agents. Our experiments show that today's leading AI systems can generally match the abilities of human novices to intermediates. At the same time, even the top models display considerable weaknesses in specific problem categories, indicating that significant work remains before they can consistently equal, or exceed, the performance of domain experts. We believe this benchmark will facilitate further research and development of LLMs and agents.

**Limitations.** Potential discrepancies between the ALE-Bench implementation and actual contest participation are discussed in Appendix A.4. In addition, this benchmark's potential limitation is the dataset size. Only 49 contests have ever been held, and the benchmark covers 40 of them. Although this is relatively small for a benchmark, each contest allows long-horizon trial and error. Because the fluctuations at individual steps are smoothed through accumulation, the variance in the outcome is low. Moreover, both organizers and the participant community trust the current ratings when evaluating humans, so we believe that 40 contests can still offer a reasonable indication of AI performance.

**Future Work.** Future work can explore two synergistic directions. First, generating synthetic problems with LLMs can expand the dataset and mitigate the risk of overfitting. This ranges from simple augmentations (e.g., altering narratives or notation) to novel challenges. While the latter complicates direct comparisons to human performance, it constitutes a rich research path. A vast synthetic problem pool could then unlock new training paradigms, like reinforcement learning, to systematically enhance open-source models. Second, ALE-Bench facilitates exploration into advanced agent architectures, including multi-agent systems with specialized roles (e.g., ideation, analysis, implementation) and multi-LLM ensembles.

**Ethical and Societal Impact.** This research is expected to accelerate progress in AI and, in particular, drive improvements in industrial optimization, such as logistics and energy scheduling. The benchmark contains no personal or sensitive data, and the experiments involve no direct human subjects. Thus, we identify no ethical concerns. We recognize that releasing this work could influence the behavior of current AHC participants. However, the work is conducted in close collaboration with the AHC organizers, and this point has been thoroughly discussed. Covert participation in AHC by AI-development teams violates the competition rules, and we strongly discourage it.

**Author Contributions.** Yuki Imajuku co-designed and implemented ALE-Bench, conducted the experiments, and led the studies. Kohki Horie designed and implemented ALE-Agent. Yoichi Iwata, Kensho Aoki, and Naohiro Takahashi assisted with the design of ALE-Bench, data preparation, and interpretation of the experimental results. Takuya Akiba initiated and managed the overall project, co-designed ALE-Bench, and provided overall guidance and supervision.

## Acknowledgements

The authors would like to thank Yotaro Kubo, Masanori Suganuma, Yutaro Yamada, and Artem Zhivolkovskiy for their helpful feedback on an earlier draft of this article.

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

# A  ALE-Bench Details

This section provides detailed information regarding the construction of the ALE-Bench dataset and the implementation of its benchmark framework. Corresponding to Section 3 in the main text, Appendix A.1 details the dataset construction process and the data provided. Appendix A.2 details the benchmark implementation, and Appendix A.3 details the evaluation metrics.

## A.1  Dataset Construction Details

From 49 contests eligible for Heuristic Rating calculation on AtCoder[1] held up to May 1, 2025, we selected 40 problems originally created by AtCoder and publicly released their data on Hugging Face.[2] Table A1 provides a comprehensive list of these problems, including their duration, genre, and top-level solution algorithms. Furthermore, one of the authors, an AtCoder Heuristic Contest (AHC) administrator, curated a *"lite"* version comprising 10 problems. These problems were selected to have relatively high difficulty and cover a diverse range of genres, also detailed in Table A1. The full version is designed to enable high-fidelity comparison with human performance, while the lite version aims to facilitate straightforward comparison among AI agents.

The data provided for each problem consists of the following components:

**Problem**  The problem statement presented to the user is provided in Markdown format. If images are used in the problem statement, these are also provided. Since AtCoder supports both English and Japanese, statements in both languages are included. If sample inputs and outputs are provided within the problem statement, these are also included. Example problem statement is shown at Figure A1.

**Scorer**  A tool for evaluating solution programs is provided. It is implemented in Rust and provided as source code. While contest participants typically cannot view the source code of the official scorer, a significant portion of its functionality is replicated by the Rust-based visualizer tool (described in the next item). Therefore, the provision of this Scorer source code is considered to have minimal impact on the fairness of the benchmark.

**Visualizer**  Separate from the problem statement, two types of tools are provided to users for visualizing the execution results of their solutions. One is a Rust-based tool executable on a local PC, which generates a static image visualizing the output of a solution program for a given input test case. (Note: Some problems, e.g., `ahc016`, do not provide this specific Rust tool.) The other is a web browser-based visualizer, enabling richer and more interactive visualizations than the Rust tool. Figure A2 shows an example. Both visualizers can also generate input test cases from random seeds and hyperparameters. Notably, the test cases for both public and private evaluation are generated using these same visualizer tools, ensuring consistency.

**Leaderboard**  Tabular data is provided to determine an equivalent rank in the original contest and calculate performance metrics based on the scores computed by the Scorer in the private evaluation. Unlike live AHCs, which feature a real-time leaderboard during the contest period, this benchmark does not offer this functionality. Actual contest participants can view other users' scores in real-time, and some might infer solution approaches from these scores. This real-time data is not included in the benchmark due to its confidential nature. Consequently, AI participants operate under slightly more challenging conditions than human contestants who would have access to this information.

In addition to problem-specific data, we provide a global leaderboard to contextualize an agent's rating within the distribution of human player ratings, specifically for determining percentile rankings. This leaderboard lists the ranks and ratings of 6,139 active users, captured at the conclusion of the `ahc046` contest, the most recent contest included in this benchmark.

---

[1] `https://atcoder.jp/contests/archive?category=0&keyword=&lang=en&ratedType=4`  (Retrieved: May 15, 2025)

[2] `https://huggingface.co/datasets/SakanaAI/ALE-Bench` (Retrieved: May 15, 2025)

## Problem Statement

AtCoder Inc. operates a food delivery service, AtCoder Foods, that leisurely delivers food that tastes good even if it gets cold. This service receives a large number of delivery orders in advance, and processes multiple deliveries simultaneously to improve efficiency. The current service area is represented as a square area $\{(x, y) \mid 0 \le x, y \le 800\}$ on a two-dimensional plane, with AtCoder's office located at the center $(400, 400)$. There are 1000 orders today, and the $i$ ($1 \le i \le 1000$)-th order is a food delivery request from a restaurant in $(a_i, b_i)$ to a location in $(c_i, d_i)$.

Today's quota for Takahashi, a delivery man, is to process 50 orders. He can freely choose a subset $S \subseteq \{1, \cdots, 1000\}$ of size exactly 50 from the 1000 orders and deliver on a route $(x_1, y_1), \cdots, (x_n, y_n)$ satisfying the following conditions.

1. For each $i \in S$, visit $(c_i, d_i)$ after visiting $(a_i, b_i)$. That is, there exists an integer pair $(s, t)$ such that $(x_s, y_s) = (a_i, b_i)$, $(x_t, y_t) = (c_i, d_i)$, and $s < t$.

2. $(x_1, y_1) = (x_n, y_n) = (400, 400)$.

After picking up food at one restaurant, he may pick up food at another restaurant or deliver food to another destination before delivering that food to the destination. He is so powerful that he can carry arbitrary numbers of dishes simultaneously.

Moving from $(x_i, y_i)$ to $(x_{i+1}, y_{i+1})$ takes time equal to the Manhattan distance $|x_i - x_{i+1}| + |y_i - y_{i+1}|$, and the total travel time for the delivery route is $T = \sum_{i=1}^{n-1} |x_i - x_{i+1}| + |y_i - y_{i+1}|$. Please optimize $S$ and delivery routes so that the total travel time is as short as possible.

## Scoring

For the total travel time $T$ of the output delivery route, you will get a score of $\mathrm{round}(10^8/(1000 + T))$. There are 100 test cases, and the score of a submission is the total score for each test case. If you get a result other than AC for one or more test cases, the score of the submission will be zero. The highest score obtained during the contest time will determine the final ranking, and there will be no system test after the contest. If more than one participant gets the same score, the ranking will be determined by the submission time of the submission that received that score.

## Input

Input is given from Standard Input in the following format:

$$a_1 \ b_1 \ c_1 \ d_1$$
$$\vdots$$
$$a_{1000} \ b_{1000} \ c_{1000} \ d_{1000}$$

Each $a_i, b_i, c_i, d_i$ is an integer between 0 and 800, inclusive, where $(a_i, b_i)$ represents the coordinates of the restaurant, and $(c_i, d_i)$ represents the coordinates of the destination. $(a_i, b_i) \ne (c_i, d_i)$ is satisfied, but for different orders $j$, there is a possibility that $\{(a_i, b_i), (c_i, d_i)\} \cap \{(a_j, b_j), (c_j, d_j)\} \ne \emptyset$.

## Output

Let the set of chosen orders be $r_1, \cdots, r_m$ ($1 \le r_i \le 1000$), and the delivery route be $(x_1, y_1), \cdots, (x_n, y_n)$ ($0 \le x_i, y_i \le 800$), output to Standard Output in the following format.

$$m \ r_1 \ \cdots \ r_m$$
$$n \ x_1 \ y_1 \ \cdots \ x_n \ y_n$$

You may output multiple times for visualization purposes. If your program outputs multiple times, only the last output will be used for scoring. The final output must satisfy $m = 50$, but intermediate outputs with $m \ne 50$ are allowed for visualization.

## Input Generation

Let $\mathrm{rand}(L, U)$ be a function that generates a uniform random integer between $L$ and $U$, inclusive. For each $i = 1, \cdots, 1000$, we generate an order $(a_i, b_i, c_i, d_i)$ as follows. We generate $a_i = \mathrm{rand}(0, 800)$, $b_i = \mathrm{rand}(0, 800)$, $c_i = \mathrm{rand}(0, 800)$, and $d_i = \mathrm{rand}(0, 800)$. Redo the generation as long as the Manhattan distance $|a_i - c_i| + |b_i - d_i|$ is less than 100.

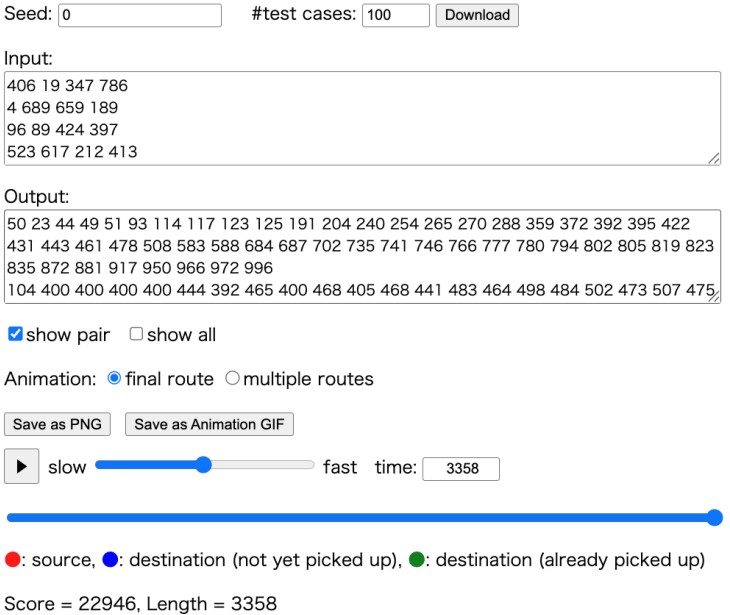

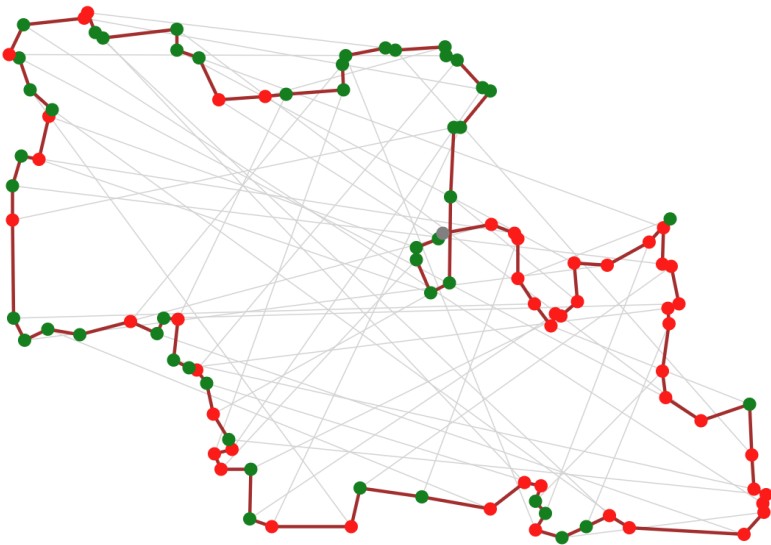

Figure A2: **The web browser-based visualization tool for** `ahc006`**.**

## A.2 Benchmark Implementation Details

We have implemented the benchmark environment as a Python library for evaluating AI agents, utilizing the dataset described above. The complete source code is publicly available via GitHub.[3]

---

[3]`https://github.com/SakanaAI/ALE-Bench` (Retrieved: May 15, 2025)

Table A1: **The list of problems in ALE-Bench.** We offer 40 problems for the full version and 10 problems for the lite version. The original contest webpage can be accessed via URLs in the format `https://atcoder.jp/contests/`*Problem ID* (e.g. `https://atcoder.jp/contests/ahc001`).

| Problem ID | Format | Hours | Judge Type | Top-Level Solution[*1] | Genre[*2] | Lite Ver. |
|---|---|---|---|---|---|---|
| `ahc001` | Long | 200 | Standard | SA | Packing | |
| `ahc002` | Short | 4 | Standard | SA | Routing | |
| `ahc003` | Long | 200 | Reactive | Bayes | Inference | |
| `ahc004` | Short | 6 | Standard | Adhoc | Packing | |
| `ahc005` | Short | 4 | Standard | SA | Routing | |
| `future-contest-2022-qual` | Long | 240 | Reactive | Bayes, Evaluation | Scheduling | |
| `ahc006` | Short | 4 | Standard | SA | Routing | |
| `ahc007` | Short | 4 | Reactive | Adhoc | Network | |
| `ahc008` | Long | 343 | Reactive | Adhoc, Structure | Game | ✓ |
| `ahc009` | Short | 4 | Standard | SA, DP | Routing | |
| `ahc010` | Short | 4 | Standard | SA | Network | |
| `ahc011` | Long | 199 | Standard | Beam, SA | Puzzle | ✓ |
| `ahc012` | Short | 4 | Standard | SA, Structure | Partitioning | |
| `ahc014` | Long | 340 | Standard | SA | Puzzle | |
| `ahc015` | Short | 4 | Reactive | Playout, Evaluation | Game | ✓ |
| `ahc016` | Long | 216 | Reactive | Bayes, Structure | Inference | ✓ |
| `ahc017` | Long | 200 | Standard | SA | Network | |
| `ahc019` | Long | 367 | Standard | Beam, SA | Puzzle | |
| `ahc020` | Short | 4 | Standard | SA | Covering | |
| `ahc021` | Short | 4 | Standard | Beam | Planning | |
| `toyota2023summer-final` | Short | 3.5 | Reactive | Playout, Evaluation | Game | |
| `ahc024` | Short | 4 | Standard | SA | Network | ✓ |
| `ahc025` | Long | 199 | Reactive | Adhoc, Bayes | Partitioning | ✓ |
| `ahc026` | Short | 4 | Standard | Beam, Evaluation | Planning | ✓ |
| `ahc027` | Long | 216 | Standard | SA, Structure | Routing | ✓ |
| `ahc028` | Short | 4 | Standard | DP, SA | Routing | |
| `ahc030` | Long | 240 | Reactive | Bayes, SA | Inference | |
| `ahc031` | Long | 240 | Standard | SA, Structure | Partitioning | |
| `ahc032` | Short | 4 | Standard | Beam | Planning | |
| `ahc033` | Long | 240 | Standard | Beam | Planning | |
| `ahc034` | Short | 4 | Standard | Flow | Planning | |
| `ahc035` | Short | 4 | Reactive | Evaluation | Game | |
| `ahc038` | Long | 240 | Standard | Beam, Structure | Planning | |
| `ahc039` | Short | 4 | Standard | SA | Partitioning | ✓ |
| `ahc040` | Long | 240 | Reactive | Bayes, Beam | Packing | |
| `ahc041` | Short | 4 | Standard | SA | Partitioning | |
| `ahc042` | Short | 4 | Standard | Beam | Planning | |
| `ahc044` | Short | 4 | Standard | SA | Scheduling | |
| `ahc045` | Long | 240 | Reactive | Bayes, SA | Network | |
| `ahc046` | Short | 4 | Standard | SA, Beam | Planning | ✓ |

[*1] SA (Simulated Annealing including hill climbing), Beam (Beam Search including greedy approaches), Bayes (Bayesian inference), Adhoc (Ad-hoc approaches), DP (Dynamic Programming), Structure (Fixed structure assumption for large solution spaces), Playout (Playout using random/greedy strategies), Flow (Network flow), Evaluation (Crafting good evaluation functions)

[*2] Covering (finding coverage methods), Routing (finding paths), Partitioning (finding partition methods), Game (dynamic game problems), Puzzle (static game problems), Scheduling (determining order), Planning (finding action sequences), Packing (finding packing methods), Network (finding connectivity), Inference (inference problems)

### A.2.1 Code Sandbox

The Code Sandbox, which replicates AtCoder's user script execution environment, was implemented using Docker.[4] The corresponding `Dockerfiles` are available in the GitHub repository.[5] We also distribute Docker Images via Docker Hub.[6] While AtCoder's execution environment evolves, this benchmark, considering the contest periods, offers two environment versions: Ver. 201907[7]

---

[4] `https://www.docker.com/` (Retrieved: May 15, 2025)

[5] `https://github.com/SakanaAI/ALE-Bench/tree/main/dockerfiles` (Retrieved: May 15, 2025)

[6] `https://hub.docker.com/r/yimjk/ale-bench` (Retrieved: May 15, 2025)

[7] `https://atcoder.jp/contests/language-test-202001` (Retrieved: May 15, 2025)

and Ver. 202301.[8] We selected programming languages with high user adoption: C++17 (GCC), Python 3 (CPython), and Rust (rustc) for Ver. 201907; and C++17 (GCC), C++20 (GCC), C++23 (GCC), Python 3 (CPython), and Rust (rustc) for Ver. 202301. We provide execution environments that replicate these setups by installing the respective programming languages and their common external libraries. However, minor discrepancies in installation procedures or specific versions of languages and libraries may exist compared to the original AtCoder environments. Crucially, as per the regulations detailed in Appendix A.3.3, the benchmark mandates the use of the Ver. 202301 environment for all problems. This standardization allows agents to utilize a consistent, modern environment even for problems from older contests. This decision was made because ecause we confirmed potential discrepancies arising from using a newer, unified environment, is small and not critical. Each execution within the sandbox is allocated resources equivalent to 1 CPU and 2GiB of RAM.

### A.2.2 Actions

We describe the specific specifications for the four types of actions introduced in Section 3.2. As previously mentioned, we refer to the benchmark execution environment where the AI participates as an `Session`.

**1. Problem** Data related to the problem can be accessed at any time once an `Session` has commenced. This includes not only the problem statement and any accompanying images, but also crucial metadata: the optimization objective (maximization or minimization of the Scorer's score), whether the problem is reactive (requiring interaction with the judging system), the problem title, original contest date and time, problem-specific time limits, execution time limits, and memory usage limits. However, as stipulated in the regulations detailed in Appendix A.3.3, some data, such as information pertaining to private evaluation test cases, must not be accessed.

**2. Test Run** AI agents can execute their generated code at any point within the problem's time limit. The framework automatically invokes the Code Sandbox for execution. When the code is executed, the source code is first compiled, and then the program runs. If compilation fails, feedback including the standard error output is provided. Successful execution yields feedback comprising the Scorer's judging result and score, the input case used, program outputs (standard output and error), execution time, and memory usage. The judging results can indicate: `COMPILATION_ERROR` for compilation failure, `MEMORY_LIMIT_EXCEEDED` for exceeding memory limits, `TIME_LIMIT_EXCEEDED` for exceeding time limits, `RUNTIME_ERROR` for runtime errors, `INTERNAL_ERROR` for internal errors within ALE-Bench not attributable to the program produced by the AI system, `WRONG_ANSWER` for incorrect answer format, or `ACCEPTED` for a correctly formatted answer with a calculated score. Crucially, the precise internal specifications of AtCoder's execution environment (e.g., judging flow, exact time/memory tracking mechanisms) are not public. Our Code Sandbox is a replication based on publicly available information (e.g., language execution commands, common libraries, machine specifications). Consequently, while the sandbox behavior is highly likely to be nearly identical to AtCoder's, exact replication in every instance cannot be guaranteed. Beyond code execution, agents can also generate new input test cases using a random seed and optional hyperparameters. The interface supports four distinct test run operations: (1) **Public Evaluation**: execution against a predefined set of test cases (50 for the full version, 5 for the lite version). (2) **Custom Test Run**: execution using an agent-specified input case. (3) **Input Case Generation**: generation of a new input case. (4) **Generate and Run**: generation of a new input case followed immediately by its execution. While public evaluation facilitates iterative testing on fixed cases, other operations allow agents to experiment with custom-generated or specified inputs. For each problem, its input cases are generatable from a single random seed. Some problems allow further customization via optional hyperparameters; agents must consult the generation tool's README to identify and utilize these.

**3. Visualization** Agents can visualize their solution's behavior on specific input cases. Similar to actual AHCs, participants have access to two types of visualizers: a simple Rust-based tool for local execution, and a more feature-rich web-based tool supporting animations

---

[8]`https://atcoder.jp/contests/language-test-202301` (Retrieved: May 15, 2025)

and interactive operations. Both are provided in this benchmark. The local Rust visualizer can optionally run concurrently with a Test Run. For the web visualizer, the `Session` can optionally launch a local HTTP server automatically. Users can then access and interact with the visualization via a web browser at the designated port. Moreover, the source code for the Rust visualizer, typically provided to human contestants, is also accessible within this benchmark.

**4. Submission** The final evaluation of an agent's solution is performed via a single private evaluation. This can be invoked only once per `Session`, within the problem's time limit, and concludes the agent's interaction with that problem. For the private evaluation, the full version uses the identical set of test cases from the original contest's private leaderboard. The lite version uses the same test cases as the full version for short-duration contests (typically hundreds of test cases). For long-duration contests (thousands of test cases), the lite version uses a reduced set: the first 10% of the full version's test cases. Following execution and scoring across the private test set, a rank is calculated. Generally, the contest score is the sum of the scorer's scores on all private test cases. This total score is then used to determine a rank against the original contest leaderboard (for the full version) or a newly rebuilt laderboard with only the lite version subset of test cases (for the lite version). However, for certain problems (e.g., `ahc016`, `ahc017`, `ahc025`), ranking is based on the sum of normalized relative scores for each test case. In these instances, normalized scores are recomputed based on the agent's and all human participants' raw scores for each test case, and the final rank is determined by sorting based on the sum of these re-normalized scores. Once the rank is established, the performance is determined. While AtCoder's official performance calculation [18] incorporates the AI system's current rating, other users' internal ratings, and the AI system's rank [19], our benchmark approximates performance based solely on the agent's equivalent rank. If ties existed in the original contest, leading to a non-unique mapping between rank and performance, we use linear interpolation from adjacent ranks and their performances to approximate the agent's performance. AtCoder applies a correction to performance values below 400 to ensure that they are positive, however, this benchmark does *not* apply this specific correction to the raw performance metric. The overall Rating, however, *does* incorporate this correction to maintain comparability with human ratings.

### A.2.3 Others

Sequential execution of multiple test cases (during test runs or final evaluation) can be prohibitively time-consuming. For instance, evaluating 1000 cases for a problem with a 2-second per-case limit would exceed 30 minutes. To mitigate this, users can specify the number of parallel executions at the beginning of an `Session`, significantly reducing overall runtime. As solutions are typically CPU-bound, the number of parallel executions should ideally not exceed the number of available physical CPU cores. Beyond the core framework, we provide common scripts to facilitate environment setup.[9] Terraform[10] scripts are also available, enabling one-command replication of our experimental setup on Amazon Web Services (AWS).

### A.3 Evaluation Metrics Details

For evaluating performance in ALE-Bench, two categories of metrics are employed: fine-grained metrics for each individual problem within the benchmark, and coarse-grained metrics for the benchmark as a whole.

### A.3.1 Per-problem Metrics

The following four fine-grained metrics are calculated per problem:

**Scorer score for each test case in private evaluation** Each test case is assigned a score by the Scorer, as defined in the problem statement. This score is to be either maximized or minimized, depending on the specific problem.

---

[9]`https://github.com/SakanaAI/ALE-Bench/tree/main/cloud` (Retrieved: May 15, 2025)

[10]`https://developer.hashicorp.com/terraform` (Retrieved: May 15, 2025)

**Overall private evaluation score** This is an aggregate score derived from the Scorer's evaluations across all private test cases. Typically, it is the sum of raw Scorer scores. However, certain problems utilize the sum of normalized scores per test case. Three primary normalization methods exist: (1) (Agent's Scorer score) / (Maximum Scorer score among all participants) (e.g., `ahc016`); (2) (Minimum Scorer score among all participants) / (Agent's Scorer score) (e.g., `ahc017`); and (3) Rank of the agent's Scorer score among all participants (e.g., `ahc025`). A higher value in this aggregate score corresponds to a better rank.

**Rank** Based on the overall private evaluation score, the agent is ranked relative to the original human participants of the contest.

**Performance** From the rank, a quantitative 'performance' metric, as used on AtCoder, is derived. Further details on its calculation are provided in Appendix A.2.

While not a built-in feature, leaderboards based on metrics other than overall performance (e.g., raw scores on specific problems) can be constructed to rank participating AI agents amongst themselves. The open-ended nature of these optimization problems means that AI agents can continue to compete and differentiate themselves even after surpassing top human performance levels, making it a suitable platform for ongoing AI vs. AI competition.

### A.3.2 Overall Metrics

The following three coarse-grained metrics are available for the overall benchmark:

**Average Performance** The arithmetic mean of the performance scores achieved on each problem in the benchmark.

**Performance Distribution** On AtCoder, both the performance and the rating (described below) are associated with color tiers. These are: $\sim 399$ (Gray), $400 \sim 799$ (Brown), $800 \sim 1199$ (Green), $1200 \sim 1599$ (Cyan), $1600 \sim 1999$ (Blue), $2000 \sim 2399$ (Yellow), $2400 \sim 2799$ (Orange), and $2800 \sim$ (Red). As these color tiers are widely recognized within the AtCoder community, analyzing the distribution of an agent's performance across these tiers provides an intuitive overview of its capabilities.

**Rating** Beyond simple averaging, AtCoder employs a 'Rating' system to aggregate performance across multiple contests, serving as an overall skill indicator. The official rating calculation formula [19] is implemented within our framework. However, as detailed in Section 3.3, this rating system incorporates adjustments specifically designed for human participation patterns (e.g., not all contests are attempted). Consequently, it is less suitable for directly evaluating AI agents that are expected to attempt all benchmark problems. It is therefore included primarily to offer a point of comparison against established human performance benchmarks.

### A.3.3 Regulations

In addition to standardizing the calculation methods for evaluation metrics, we have established detailed regulations concerning the execution environment and agent actions. These regulations aim to ensure fair and consistent evaluation of AI agents, both in comparison to human contestants and other AI systems. We outline the key clauses of our bespoke regulations below:

- Permitted programming languages are C++ (versions 17, 20, or 23), Rust, and Python, all within the AtCoder Judge Version 202301 environment.

- Execution time is measured as the maximum of wall-clock time and CPU time. While parallelization within a solution is permitted, it does not reduce the officially measured execution time (as per AtCoder rules).

- Participants may use any development environment or editor of their choice.

- Use of self-created libraries and publicly available human-created libraries is permitted. However, direct reuse of code identifiable as a complete AHC submission by another participant is prohibited to ensure originality of the agent's core logic.

- General internet searches for programming concepts or libraries are allowed. However, accessing websites containing specific keywords like "AtCoder," "AHC," or the exact

problem name (which might lead to existing solutions or discussions) is forbidden. An exception is made for interacting with the local visualization pages provided by ALE-Bench.

- No human intervention or assistance is permitted after the agent has received the problem statement and commenced its run. The sole exception is for restarting the process due to critical technical failures (e.g., network outages, hardware crashes).

- The size of each submitted source code must not exceed 512 KiB.

- The time limit for each problem mirrors the duration of the original AtCoder contest. Agents are not required to use the full duration and may submit their solution earlier.

- Private evaluation can be initiated only once per problem. The agent must select a single solution code for this evaluation, and the process must start before the problem's time limit expires. For practical purposes, if a solution was fully generated and saved before the deadline, its private evaluation can be run even if the submission command itself is issued shortly after the time limit.

- Agents may utilize unlimited computational resources for development and code generation (e.g., CPUs, GPUs). However, the official ALE-Bench execution environment for scoring submissions must run on an Intel® Xeon® Platinum 8375C CPU @ 2.90GHz, or a CPU with demonstrably equivalent performance (e.g., AWS C6i series instances [17]). If non-compliant hardware is used, results can still be reported but must be accompanied by detailed specifications of the computational environment to allow for potential performance normalization or caveats.

- Agents must not access or utilize any information pertaining to: (a) the private evaluation test cases themselves (beyond what is inferable from the public visualizer/generator), (b) the raw scores or detailed data of individual human participants on the leaderboard (used for normalization in some problems).

## A.4 Limitations of ALE-Bench

ALE-Bench is designed to reproduce actual contests so that humans and AI can be compared fairly. Nevertheless, it has the following limitations: *(1) Differences in judging environments.* Contests held before the summer of 2023 ran on an older system whose hardware is slightly slower than today's environment. However, even when we re-evaluated some solutions with the latest environment, their scores and rankings showed little change, so we can conclude that the impact is minor. *(2) Use of new resources.* Even the earliest contest (March 2021) is relatively recent. Yet, algorithms or implementations that were unavailable then can now be applied, allowing higher performance to be achieved more easily than was originally possible. *(3) Problem-set contamination.* Since the original problems are publicly available, they may have been accidentally included in AI training data. According to the results in Section 5.4, no contamination effects have been detected, but we will continue adding fresh problems in future updates and monitor the situation.

# B    ALE-Agent Details

This section details the conceptual algorithmic framework and strategic design underpinning ALE-Agent's solution exploration capabilities.

## B.1    Core Strategy and Design Philosophy

ALE-Agent achieves efficient exploration of solutions for heuristic problems by integrating the broad knowledge and code generation prowess of Large Language Models (LLMs) with a systematic search algorithm. Our approach utilizes an algorithm based on best-first search, striving for a balance between the diversity of LLM-generated solutions and the depth of the search. This process emulates human-like trial-and-error, where diverse LLM-generated ideas are quantitatively evaluated, and the most promising solution candidates are further explored in depth.

## B.2    Search Algorithm

ALE-Agent's search algorithm expands a search tree based by best-first search principles. Each node in the search tree represents a state, each holding its corresponding source code. In each iteration, the

algorithm selects the most 'promising' state from the current frontier of explored states. The source code of this selected state then serves as a seed for generating a new cohort of candidate solutions through LLM-driven refinement techniques (detailed in Appendix B.3). The "promisingness" of a state is quantified through local execution on a fixed suite of 50 test cases, employing a composite scoring function based on:

1. **Acceptance Ratio**: The proportion of test cases for which the generated code is `ACCEPTED`. A higher acceptance ratio indicates a more promising solution.
2. **Score**: Among `ACCEPTED` solutions, those achieving a better problem-specific score (as defined by the contest) are considered more promising.

The selection of the next node for expansion is guided by a priority score that holistically integrates these two criteria.

Diverging from standard best-first search, which expands a single successor, ALE-Agent adopts a beam search-inspired strategy, concurrently generating a beam of $k = 30$ child nodes from each selected parent node. This strategy enhances solution diversity and optimizes parallel processing efficiency (detailed in Appendix B.4). Furthermore, we incorporate a tabu search-like mechanism: previously expanded parent nodes are precluded from re-selection for expansion. These enhancements preserve the core efficiency of best-first search while substantially improving exploration breadth and reducing susceptibility to premature convergence on local optima.

## B.3   State Transition

Node expansion, corresponding to the generation of novel solution candidates, is orchestrated through an interactive dialogue with the LLM. In this dialogue, the agent provides the LLM with comprehensive context about the current solution (e.g. its source code, past evaluation results, and any evaluative feedback). Then the LLM is prompted to generate either refinements to the existing code or entirely new algorithmic approaches.

### B.3.1   Initial Solution Generation

To generate initial solutions (i.e., children of the search tree's root node), the LLM is provided with the complete problem description. Based on this, the LLM formulates one or more initial algorithmic approaches and generates the corresponding source code. This approach eliminates the need for human-seeded initial solutions, allowing the search to commence from a diverse set of starting points derived purely from the LLM's understanding of the problem.

### B.3.2   Iterative Solution Refinement

Generating successor solutions from an existing state involves an iterative refinement loop. In each iteration, the LLM is prompted with a rich contextual package comprising:

(a) **Current Code and Performance Feedback:** The source code of the current solution and its detailed performance feedback (e.g., per-test-case scores, execution status), which serves as the primary basis for LLM-driven refinement.

(b) **Historically Best Code and Performance Feedback:** The source code of the best-performing solution discovered so far in the search, along with its associated feedback. This helps to prevent the search from stagnating in suboptimal regions of the solution space by reminding the LLM of previously successful strategies.

(c) **Summary of Past Search Trajectory:** A concise summary of recent search history (e.g., a few previous attempts, their outcomes, and applied modifications). This enables the LLM to engage in more informed, longer-range strategic planning by considering the evolution of explored strategies, their efficacy, and identified pitfalls.

(d) **Targeted Improvement Guidance:** To encourage diverse exploration and steer the LLM towards promising avenues, specific improvement directives are stochastically selected from a predefined set of guiding prompts. These prompts might suggest, for instance, optimizing a particular algorithm (e.g., simulated annealing or beam search), or speeding up the solution by introducing new algorithms or data structures. This aims to scaffold the LLM's reasoning towards discovering solutions well-suited for heuristic problems.

Based on this multifaceted input, the LLM first formulates a high-level improvement strategy. It then attempts to implement this strategy by generating new source code over a sequence of three interactive conversational turns. The code version exhibiting the best performance (evaluated locally) across these three turns is selected as the basis for a new node in the search tree. This multi-turn refinement protocol provides the LLM with opportunities for self-correction and allows for a more nuanced implementation of its proposed strategic changes.

## B.4 Parallel Execution for Enhanced Throughput

ALE-Agent leverages parallel processing when generating multiple child nodes from a single parent to maximize search throughput. Although reasoning models, such as OpenAI's o-series or Google's Gemini 2.5, may exhibit response latencies of up to several minutes, their API calls are readily parallelizable. ALE-Agent dispatches parallel requests to the reasoning model, where each task prompts the model to generate a code refinement based on the parent node's information. Upon receiving responses from the model, the newly generated code segments are added to an evaluation queue for asynchronous assessment. Our local evaluation pipeline, when assessing a solution against 50 input cases using 13 parallel threads, takes approximately 10 seconds per code instance. By generating 30 child nodes concurrently at each expansion step, ALE-Agent effectively utilizes local computational resources for evaluation and minimizes the impact of the reasoning model latency on overall search velocity.

## B.5 Ablation Configurations

To facilitate ablation studies, ALE-Agent can be configured in several operational modes:

**Base:** This is the most fundamental setting, which omits tree search (i.e., beam width of 1) and the explicit domain-guided improvement directives. Its features are:

1. It employs a sequential refinement strategy, generating only one child node from each parent.
2. When implementing an improvement strategy, it forgoes the three-turn iterative refinement; code is adopted as a successor as soon as it passes all local test cases.
3. It does not use the targeted improvement guidance prompts (described in item (d) of Appendix B.3.2).

**+ Method 1:** This mode builds upon **Base** by reintroducing (i) the three-turn iterative refinement for code implementation and (ii) the targeted improvement guidance prompts, while still maintaining the single-path search (beam width 1). These feature enable LLMs to implement potentially complex, domain-informed strategies, without introducing complex tree search.

**+ Method 1&2:** This configuration represents the full ALE-Agent but with one specific modification: it extends **+ Method 1** by employing the full beam search mechanism (beam width of 30). However, the summary of past search trajectory (item (c) of the iterative refinement context in Appendix B.3.2) is omitted, as this history might inadvertently reduce diversity when multiple distinct paths are being explored concurrently in the tree search. This allows the agent to maximize the effectiveness of tree search.

## B.6 Prompts for guided improvement

We provide the prompts for the targeted improvement guidance in Appendix B.3.2 here. We use the following four prompts reflecting the domain knowledge:

```
Speedup Prompt

Based on the code and feedback: What are the key algorithms and data structures
    used? What are its computational complexity bottlenecks (consider feedback
    like Time Limit Exceeded)? How might we improve the time or space complexity
    (consider feedback like Memory Limit Exceeded)? Consider both small
    optimizations and completely different approaches. You don't need to
    implement the solution yet. Instead, please think deeply and broadly about
    the possible improvements, and explain your thoughts.
```

**Simulated Annealing State Improvement Prompt**

```
If this solution uses simulated annealing, analyze the problem, solution
    properties, and feedback to suggest better state representation. Consider how
     the current state encoding might be limiting the search space or convergence
     speed, especially in light of the feedback. Think about alternative state
    encodings that could lead to better local optima or faster convergence. You
    don't need to implement the solution yet. Instead, please think deeply and
    broadly about the possible improvements, and explain your thoughts.
```

**Simulated Annealing Neighbor Improvement Prompt**

```
If this solution uses simulated annealing, analyze the problem, solution
    properties, and feedback to suggest better neighborhood design. Consider how
    the current neighborhood structure might be limiting the search space
    exploration or convergence speed, based on the feedback. Think about
    alternative neighborhood structures that could lead to better local optima or
     faster convergence. Specifically, consider:
1. How to balance between small and large moves in the search space
2. How to ensure the neighborhood structure allows reaching any valid solution
3. How to design moves that maintain solution feasibility while exploring new
    regions
You don't need to implement the solution yet. Instead, please think deeply and
    broadly about the possible improvements, and explain your thoughts.
```

**Beam Search Improvement Prompt**

```
Consider implementing or improving a beam search approach. Think about the beam
    width and evaluation function that could lead to better solutions. Consider
    how to effectively balance between diversity and quality in your beam. You
    don't need to implement the solution yet. Instead, please think deeply and
    broadly about the possible improvements, and explain your thoughts.
```

## C   Additional Experimental Details and Results

### C.1   Experimental Setup Details

**Computational Resource.**   All experiments were conducted on Amazon EC2 `c6i.32xlarge` [17] instances, each equipped with 128 vCPUs and 256 GiB of RAM. These instances feature Intel® Xeon® Platinum 8375C CPUs @ 2.90GHz, identical to those used in the AtCoder execution environment (Ver. 202301). The operating system was `Ubuntu 22.04.5 LTS` with Linux kernel `6.8.0-1027-aws x86_64`. All instances were located in the AWS `us-east-1` (N. Virginia) region[11]. No GPUs were utilized in these experiments. Python 3.12.9 served as the interpreter for all experimental tasks. Each instance concurrently processed a maximum of four distinct problems, with these concurrent tasks sharing all available instance resources. As detailed in Appendix A.2, ALE-Bench allows parallel execution of multiple test cases for a single problem. We configured it to run up to 13 test cases in parallel per problem. Given that the evaluated programs were expected to be CPU-bound with minimal I/O latency, this configuration (up to 4 problems $\times$ 13 cases/problem = 52 concurrent processes if fully utilized) was well within the 64 physical cores of the `c6i.32xlarge` instance, ensuring ample computational resources.

**Models.**   We experimented with a total of 22 leading LLMs. More specifically, we used five non-reasoning models (GPT-4o-mini [20], GPT-4o [21], GPT-4.1-nano [22], GPT-4.1-mini [22], and GPT-4.1 [22]) and four reasoning models (o1 [23], o3-mini [24], o3 [25], and o4-mini [25]) from OpenAI; five non-reasoning models (Gemini 1.5 Flash-8B [26], Gemini 1.5 Flash [26], Gemini 1.5 Pro [26], Gemini 2.0 Flash-Lite [27], and Gemini 2.0 Flash [28]) and two reasoning models (Gemini

---

[11]https://docs.aws.amazon.com/global-infrastructure/latest/regions/aws-regions.html (Retrieved: May 15, 2025)

Table A2: **The details of LLMs with API call.**

| Model | API Provider | Model Name | Parameters |
|---|---|---|---|
| GPT-4o mini | OpenAI | gpt-4o-mini-2024-07-18 | |
| GPT-4o | OpenAI | gpt-4o-2024-08-06 | |
| GPT-4.1 nano | OpenAI | gpt-4.1-nano-2025-04-14 | |
| GPT-4.1 mini | OpenAI | gpt-4.1-mini-2025-04-14 | |
| GPT-4.1 | OpenAI | gpt-4.1-2025-04-14 | |
| o1-high | OpenAI | o1-2024-12-17 | reasoning_effort:"high" |
| o3-mini-high | OpenAI | o3-mini-2025-01-31 | reasoning_effort:"high" |
| o3-high | OpenAI | o3-2025-04-16 | reasoning_effort:"high" |
| o4-mini-high | OpenAI | o4-mini-2025-04-16 | reasoning_effort:"high" |
| Gemini 1.5 Flash-8B | Google AI Studio | gemini-1.5-flash-8b-001 | |
| Gemini 1.5 Flash | Google AI Studio | gemini-1.5-flash-002 | |
| Gemini 1.5 Pro | Google AI Studio | gemini-1.5-pro-002 | |
| Gemini 2.0 Flash-Lite | Google AI Studio | gemini-2.0-flash-lite-001 | |
| Gemini 2.0 Flash | Google AI Studio | gemini-2.0-flash-001 | |
| Gemini 2.5 Flash | Google AI Studio | gemini-2.5-flash-preview-04-17 | |
| Gemini 2.5 Pro | Google AI Studio | gemini-2.5-pro-preview-03-25 | |
| Claude 3.5 Haiku | AWS Bedrock | us.anthropic.claude-3-5-haiku-20241022-v1:0 | max_tokens:8192 |
| Claude 3.5 Sonnet | AWS Bedrock | us.anthropic.claude-3-5-sonnet-20241022-v2:0 | max_tokens:8192 |
| Claude 3.7 Sonnet | OpenRouter (Google/Anthropic) | anthropic/claude-3.7-sonnet | max_tokens:20000 |
| Claude 3.7 Sonnet (Thinking) | OpenRouter (Google/Anthropic) | anthropic/claude-3.7-sonnet | max_tokens:20000, thinking_budget_tokens:16000 |
| DeepSeek-V3 | OpenRouter (Lambda/DeepInfra) | deepseek/deepseek-chat-v3-0324 | temperature:0.3, max_tokens:8000 |
| DeepSeek-R1 | OpenRouter (Lambda/DeepInfra) | deepseek/deepseek-r1 | temperature:0.6, max_tokens:32768 |

2.5 Flash [29] and Gemini 2.5 Pro [30]) from Google; three non-reasoning models (Claude 3.5 Haiku [31], Claude 3.5 Sonnet [31], and Claude 3.7 Sonnet [32]) and one reasoning model (Claude 3.7 Sonnet (Thinking) [32]) from Anthropic; and one non-reasoning model (DeepSeek-V3 [33]) and one reasoning model (DeepSeek-R1 [34]) from DeepSeek. All models were accessed via their publicly available APIs. Table A2 lists the API provider and parameter settings used for each model. For models accessed via OpenRouter, the underlying provider is indicated in parentheses. When using OpenHands, the effective model parameters might differ from those listed in Table A2, as OpenHands' internal configuration takes precedence.

**Prompts.** A consistent set of prompt templates was used across all experiments, unless otherwise specified. The model was provided with the problem statement, feedback from the ALE-Bench environment on previously generated code (if applicable), and an instruction to generate code within a designated Markdown-style code block. The solution code was then extracted from the model's response using regular expression pattern matching. Images accompanying the problem statements were not used.

In the prompt templates, placeholders denoted by `${}` are substituted with either the content described in the surrounding sentences or the values of the corresponding variables listed below:

**TIME_LIMIT:** The execution time limit in seconds provided by each problem. (e.g. `2.0`)

**MEMORY_LIMIT:** The memory limit in MiB provided by each problem. (e.g. `1024`)

**PROBLEM_STATEMENT:** The English problem statement in Markdown format provided for each problem.

**CODE_LANGUAGE_NAME:** `C++20 (gcc 12.2.0)` for C++20, `Python (CPython 3.11.4)` for Python3, `Rust (rustc 1.70.0)` for Rust.

**CODE_BLOCK:** ```` ```cpp\n// Your code here\n``` ```` for C++20, ```` ```python\n# Your code here\n``` ```` for Python3, ```` ```rust\n// Your code here\n``` ```` for Rust.

**EXTERNAL_LIBRARIES:** External libraries that users can use in the AtCoder execution environment.

```
External Libraries (C++20)

 - AC Library@1.5.1
 - Boost@1.82.0
 - GMP@6.2.1
 - Eigen@3.4.0-2ubuntu2
```

## External Libraries (Python3)

```
- numpy==1.24.1
- scipy==1.10.1
- networkx==3.0
- sympy==1.11.1
- sortedcontainers==2.4.0
- more-itertools==9.0.0
- shapely==2.0.0
- bitarray==2.6.2
- PuLP==2.7.0
- mpmath==1.2.1
- pandas==1.5.2
- z3-solver==4.12.1.0
- scikit-learn==1.2.0
- ortools==9.5.2237
- ac-library-python
- setuptools==66.0.0
- cppyy==2.4.1
- torch==1.13.1
- polars==0.15.15
- lightgbm==3.3.1
- gmpy2==2.1.5
- numba==0.57.0
```

## External Libraries (Rust)

```
- ac-library-rs@=0.1.1
- once_cell@=1.18.0
- static_assertions@=1.1.0
- varisat@=0.2.2
- memoise@=0.3.2
- argio@=0.2.0
- bitvec@=1.0.1
- counter@=0.5.7
- hashbag@=0.1.11
- pathfinding@=4.3.0
- recur-fn@=2.2.0
- indexing@=0.4.1
- amplify@=3.14.2
- amplify_derive@=2.11.3
- amplify_num@=0.4.1
- easy-ext@=1.0.1
- multimap@=0.9.0
- btreemultimap@=0.1.1
- bstr@=1.6.0
- az@=1.2.1
- glidesort@=0.1.2
- tap@=1.0.1
- omniswap@=0.1.0
- multiversion@=0.7.2
- num@=0.4.1
- num-bigint@=0.4.3
- num-complex@=0.4.3
- num-integer@=0.1.45
- num-iter@=0.1.43
- num-rational@=0.4.1
- num-traits@=0.2.15
- num-derive@=0.4.0
- ndarray@=0.15.6
- nalgebra@=0.32.3
- alga@=0.9.3
- libm@=0.2.7
- rand@=0.8.5
- getrandom@=0.2.10
- rand_chacha@=0.3.1
- rand_core@=0.6.4
- rand_hc@=0.3.2
- rand_pcg@=0.3.1
- rand_distr@=0.4.3
- petgraph@=0.6.3
- indexmap@=2.0.0
- regex@=1.9.1
- lazy_static@=1.4.0
```

```
- ordered-float@=3.7.0
- ascii@=1.1.0
- permutohedron@=0.2.4
- superslice@=1.0.0
- itertools@=0.11.0
- itertools-num@=0.1.3
- maplit@=1.0.2
- either@=1.8.1
- im-rc@=15.1.0
- fixedbitset@=0.4.2
- bitset-fixed@=0.1.0
- proconio@=0.4.5
- text_io@=0.1.12
- rustc-hash@=1.1.0
- smallvec@=1.11.0
```

We employed seven distinct prompt templates in our experiments, as presented below:

1. **System Prompt**: Used for the system message (also known as the developer message for some OpenAI models).

2. **Initial Instruction Prompt**: The message for the first trial of code generation with the model.

3. **Feedback for No Code Block Found**: Used to instruct the model to regenerate its response when the specified code block is missing from its response.

4. **Feedback for Accepted Code**: Provided to the LLM in the subsequent turn as feedback when the public evaluation of the generated code results in an ACCEPTED status.

5. **Feedback for Non-Accepted Code**: Provided to the LLM in the subsequent turn as feedback when the public evaluation of the generated code does not result in an ACCEPTED status.

6. **Refinement Instruction Prompt**: Used from the second code generation attempt onwards. It instructs the model to refine its previous solution based on the feedback from the public evaluation of the code generated in the prior turn.

7. **Refinement Instruction Prompt with Summarization**: Similar to the Refinement Instruction Prompt, but used from the second code generation attempt onwards when summarization of the conversation history is employed. This template instructs the model to generate a new solution based on a summary of past interactions, rather than the full history.

Experiments involving scaffolding systems (OpenHands, ALE-Agent) utilized slightly modified versions of these templates, tailored to each system. Additionally, specific prompts unique to each scaffolding system were also employed, as detailed in their respective sections.

**System Prompt**

```
You are a world-class algorithm engineer, and you are very good at programming.
    Now, you are participating in a programming contest. You are asked to solve a
     heuristic problem, known as an NP-hard problem.
```

**Initial Instruction Prompt**

```
There is a problem statement at the end of this message. First, please analyze the
    problem statement. Please think about the essential points of the problem
    and possible algorithms to get higher rank in the contest. Next, please
    implement your solution in ${CODE_LANGUAGE_NAME}. Your solution code should
    be written in the ${CODE_BLOCK} code block. You can use external libraries as
     follows:
${EXTERNAL_LIBRARIES}

[Problem statement]
Execution time limit: ${TIME_LIMIT} sec / Memory limit: ${MEMORY_LIMIT} MB
${PROBLEM_STATEMENT}
```

## Feedback for No Code Block Found

```
No valid code block found. Please implement your solution in ${CODE_LANGUAGE_NAME
    }. Your solution code should be written in the ${CODE_BLOCK} code block.
```

## Feedback for Accepted Code

```
[Public test result]
Overall judge result: ACCEPTED
Overall absolute score: ${(Overall score is provided here.)}$
- Case 1: ${(Score for case 1 is provided here.)}$
- Case 2: ${(Score for case 2 is provided here.)}$
...(omission)...
- Case $N: ${(Score for case $N is provided here.)}$
```

## Feedback Template for Non-Accepted Code

```
[Public test result]
Overall judge result: ${(Judge result is provided here. It is one of these words:
    COMPILATION_ERROR, MEMORY_LIMIT_EXCEEDED, TIME_LIMIT_EXCEEDED, RUNTIME_ERROR,
     INTERNAL_ERROR, WRONG_ANSWER)}
- Case $IDX:
    Absolute score: ${Score for case $IDX in seconds here.}
    Execution time: ${Execution time in seconds for case $IDX here.} sec
    Memory usage: ${Memory usage in MiB for case $IDX here.} MB
    Standard error: "${The standard error output for case $IDX in here.}"
    Message: "${The feedback message from the ALE-Bench session for case $IDX in
        here.}"
```

## Refinement Instruction Prompt

```
${The feedback for the latest code here.}

Based on the above feedback, please consider the ways to improve your solution.
    Firstly, please analyze this given feedback and list what insights can be
    gained from it. Then, based on the insights, please refine your code to
    achieve better performance. It can be a simple bug fix, the introduction of a
     new algorithm, or any degree of change from minor to major. Your solution
    code should be written in the ${CODE_BLOCK} code block.
```

## Refinement Instruction Prompt (with Summarization)

```
[Problem statement]
Execution time limit: ${TIME_LIMIT} sec / Memory limit: ${MEMORY_LIMIT} MB
${PROBLEM_STATEMENT}

[Summary of your previous attempts]
${Summary that the model output in the previous turn here. If it is the first time
    , we used "Your first submission was done and you need to start logging your
    attempts from now on." string for the initial summary. If there is no summary
     in the previous response from the model, we used "Your summary was not found
    . The summary must be written in the Markdown format in the ``md\n<!-- Your
    summary here -->\n`` code block." string for the summary.}

[Your best submission]
### Code
${The current best code with code block template here.}

### Feedback
${The feedback for the current best code here.}

[Your latest submission]
### Code
```

```
${The latest code with code block template here. If the latest code is the same as
    the current best code, we used "The latest code is the same as the best code
    ." string.}

### Feedback
${The feedback for the latest code here. If the latest code is the same as the
    current best code, we used "The latest feedback is the same as the best
    feedback." string.}

Based on the above feedback, please consider the ways to improve your solution.
    Firstly, please analyze this given feedback and list what insights can be
    gained from it. Apart from that, please create a new summary including the
    content of the summary of your previous attempts in Markdown format in the
    ```md\nYour summary here\n``` code block. If this code block in this format
    is not found, the summary of your previous attempts will not be input in the
    next turn. Then, based on the insights, please refine your code to achieve
    better performance. It can be a simple bug fix, the introduction of a new
    algorithm, or any degree of change from minor to major. Your solution code
    should be written in the ${CODE_BLOCK} code block.
```

**Metrics.** The primary evaluation metrics, introduced in Section 3.3 and further detailed in Appendix A.3, were: (1) average performance across all benchmark problems, (2) performance distribution, and (3) AtCoder-style Rating. For the performance distribution, we report the number of problems for which each color-tier (e.g., Cyan, Red) was achieved. Additionally, we computed two auxiliary cost metrics: the average monetary cost (USD) to solve a single problem and the average cost per model response. The cost per response is not strictly equivalent to the cost per code generation, as some responses might lack a valid code block. These calculations are based solely on input and output token counts, excluding potential discounts from caching or other pricing variations. Cost calculation for models accessed via OpenAI was: (`prompt_tokens` × input token price) + (`completion_tokens` × output token price). For Google AI Studio: (`prompt_token_count` × input token price, considering volume tiers) + (`candidates_token_count` × output token price, considering input token count and thought tokens). For AWS Bedrock: (`input_tokens` × input token price) + (`output_tokens` × output token price). For OpenRouter, we used the sum of the `cost` field from its log files.

## C.2   Details of the One-Shot Experiment

**Setup.** To assess the one-shot algorithm engineering capabilities of LLMs, we conducted experiments where models had a limited number of attempts to generate a correct solution, without extensive iterative refinement. Ideally, evaluation would rely solely on the model's first response. However, the initial response might lack a code block (e.g., containing only a high-level plan), or the code might fail due to minor, non-fundamental issues such as exceeding time limits or output formatting errors. Such failures could lead to an unfairly low assessment of the model's core algorithm engineering abilities. Therefore, to mitigate these issues while preserving focus on one-shot capabilities, we allowed each model to generate up to five code submissions per problem before private evaluation. The protocol was: After each code generation, a public evaluation was performed. If the result was `ACCEPTED`, no further generation for that problem was permitted. Otherwise, feedback (per defined templates) was provided, and the model attempted to revise its code. If an `ACCEPTED` status was not achieved after five attempts, the last generated code was used for private evaluation. If a model's response lacked a code block in the specified format, it was prompted to regenerate correctly, with up to three such retries allowed. The conversation started with the *Initial Instruction Prompt*. Subsequent user messages for refinement used the *Refinement Instruction Prompt*. The *Feedback for No Code Block Found* prompt was used if a code block was missing. The full conversation history was maintained and provided to the model for each new response generation in this setting. These experiments, processing the full ALE-Bench, ran on 10 `c6i.32xlarge` instances in parallel and were completed within one day.

**Results.** Table A3 presents the complete results for each model and programming language in the one-shot setting. Rows shaded in light gray indicate the average performance for each model across the three languages (C++20, Python3, and Rust). The results reveal a clear performance disparity among the languages. ALE-Bench is designed to mirror the actual contest environment, where language choice has strategic implications. For the typically CPU-bound tasks in AHC, the advantage

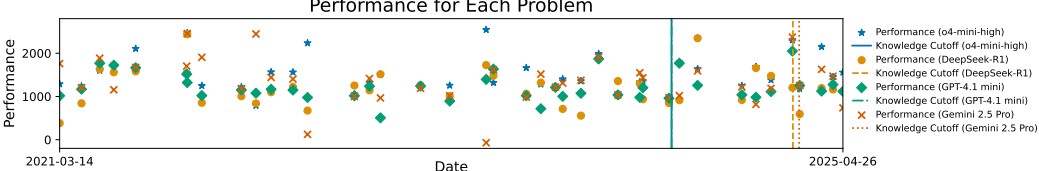

Figure A3: **Performance plot for each model.** Similar to Figure 6, For each model, a scatter plot is shown with contest end dates on the x-axis and performance on the y-axis. Each vertical line indicates the knowledge cutoff date of each model.

of native compiled languages (C++ and Rust) over Python is an expected outcome, reflecting real-world scenarios where execution speed is a crucial factor. Actually, top contestants usually use C++ or Rust. More interestingly, we observe a performance gap even between C++ and Rust, which are often considered comparable in human competitions. This suggests that our benchmark captures nuances beyond raw execution speed, potentially related to the LLM's proficiency with each language's syntax and libraries or the quality of the generated code.

### C.3    Details of the Iterative-Refinement Experiment

**Setup.**    To evaluate the algorithm engineering capabilities of LLMs with extended "thinking" time, we conducted experiments with a time limit per problem of either four hours or the problem's original contest duration, whichever was shorter. This four-hour timeframe, as indicated in Table A1, aligns with the duration of most short-format AtCoder Heuristic Contests, allowing AI systems to virtually participate under time constraints comparable to human contestants. Unlike the one-shot experiments, models in this setting continuously received public evaluation feedback and refined their solutions throughout the allocated time, even after achieving an `ACCEPTED` status, to maximize their scores. A four-hour period can result in numerous code generations (potentially ∼100). Submitting the full conversation history for each generation would exceed context limits and incur prohibitive costs. Thus, we employed a summarization strategy: after the first code generation, the LLM summarized its progress. For subsequent generations, the model received the original problem statement, its summary of previous attempts, the current best-performing code (with its public evaluation feedback), and the most recent code (with its feedback). The first user message was the *Initial Instruction Prompt*, as in the one-shot setup. Subsequent refinement messages used the *Refinement Instruction Prompt (with Summarization)*. The *Feedback for No Code Block Found* prompt was used for missing code blocks. With summarization, the explicit conversation history was cleared before each new code generation request (unless retrying for a missing code block, where the immediate prior history was retained). The final submission for each problem was the code that achieved an `ACCEPTED` status on public tests and yielded the highest total score from individual test case scorers. These iterative-refinement experiments utilized four `c6i.32xlarge` instances. Each instance was dedicated to one of four models: GPT-4.1 mini, o4-mini-high, Gemini 2.5 Pro, and DeepSeek-R1. Each model processed all 40 problems from the full ALE-Bench. On each instance, up to four problems were processed concurrently (see Appendix C.1), adhering to the specified time limit. This phase took approximately two days.

**Results.**    The main aggregated results for the iterative-refinement experiments are presented in Table 3 in the main body of the paper. Detailed per-problem performance for this set of experiments is provided in Table A6 and plotted in Figure A3.

Table A4 provides the AtCoder user statistics that were used for the performance distribution comparison with o4-mini-high, as discussed in Section 5.2 of the main paper. These statistics were computed from internal AtCoder data by one of the authors, who is an employee of AtCoder Inc.

Figure A4 provides additional examples illustrating the trajectories of public evaluation scores and the file sizes of generated code over the four-hour period. In some instances, such as o4-mini-high on problem `ahc045`, the public evaluation score showed little improvement over time. Code file sizes also varied significantly depending on the model and the problem, as can be observed in the figures.

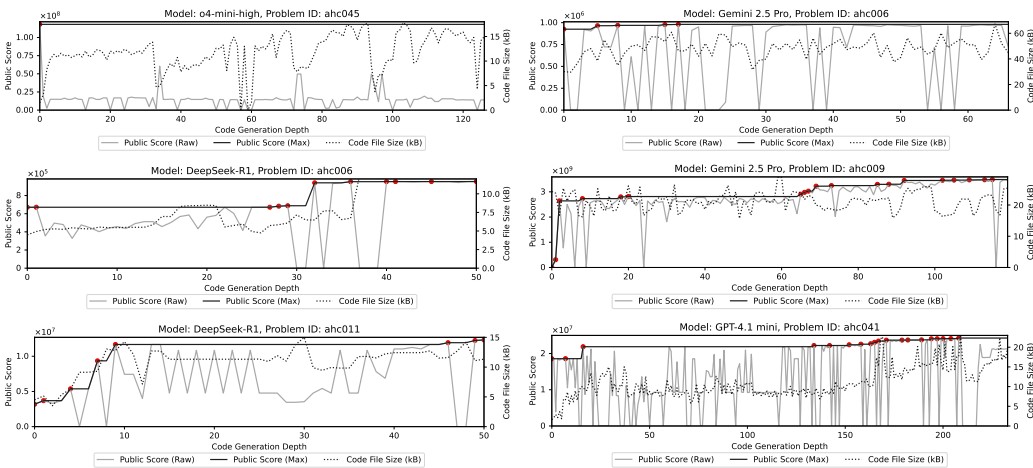

Figure A4: **Trends in public score and code file size in the iterative-refinement setting.** Each plot shows the progression of generated code file sizes alongside the corresponding public evaluation scores over a four-hour period. Points farther to the right represent that the code is generated at later time points.

Execution log analysis revealed distinct reasons for Gemini 2.5 Pro and DeepSeek-R1 scoring below the brown performance tier on one or two problems. Gemini 2.5 Pro occasionally produced solutions running very close to the time limit (e.g., 1.98s for a 2s limit). While passing most test cases (`ACCEPTED`), slight execution time fluctuations, common in competitive programming, caused a few `TIME_LIMIT_EXCEEDED` results. This led to a zero score under ALE-Bench rules, lowering overall performance. Such submissions are generally discouraged in AtCoder contests due to potential execution time variations; hence, we report these results as observed. DeepSeek-R1, conversely, exhibited issues with instruction following. For some problems, it frequently failed to generate the requested summary of its attempts. Consequently, prior work was not effectively leveraged in subsequent iterations, leading to low performance as the four-hour limit elapsed without substantial progress.

The six problems where o4-mini-high scored ≥2000 were all short-format contests amenable to simulated annealing. Other AIs also predominantly scored ≥2000 on such problems. A notable exception was DeepSeek-R1 on `ahc035`, which required designing an ad-hoc evaluation function. The `ahc035` involved creating a "seed" (game state) maximizing a sum of evaluation criteria. A key insight was to focus on each criterion's maximum possible value and preserve these during seed "cultivation." DeepSeek-R1's solution appeared to greedily select seeds with many maximum-value components and then greedily place/configure them to maximize a bit-wise OR of adjacent pairs (a problem objective). This strategy aligns with a reasonable baseline. Higher-ranking humans often refined such greedy approaches, e.g., using simulated annealing for placement or incorporating non-maximum items into scoring. DeepSeek-R1's solution implemented fundamental aspects for a decent score but lacked these advanced optimizations.

## C.4 Details of Evaluating Scaffolding Experiment

**Setup.** In this set of experiments, we compared different scaffolding systems. Specifically, we evaluated the existing OpenHands system (Version 0.34.0) [35] and our proposed ALE-Agent, which is detailed in Section 4. All experiments in this section were conducted under the same time constraints as the iterative-refinement setting: a maximum of four hours per problem or the problem's original contest time limit, whichever was shorter.

Our first scaffolding experiment evaluated the CodeActAgent from OpenHands (`v0.34.0`) on the full ALE-Bench. OpenHands allows file system interaction; solutions are typically delivered as files. We instructed the agent to save its final C++ code to `submission.cpp` for evaluation. The agent attempted each problem until the time limit or voluntary exit. The content of `submission.cpp` at termination (or timeout) was used for final evaluation. If the CodeActAgent requested user input,

we performed a public evaluation on the current `submission.cpp` and provided the feedback. The ALE-Bench C++20 Docker image served as the OpenHands code sandbox. The prompt included allowed external libraries, compilation, and execution commands. As CodeActAgent supports browser interaction, we ran the ALE-Bench local visualization server. Feedback to the agent included the visualizer's port and instructions for data input. The tailored prompts for OpenHands are provided below. The *Initial Instruction Prompt (OpenHands)* initiated tasks. If user input was requested, the *Refinement Instruction Prompt (OpenHands)* was used. For public evaluation feedback, the standard *Feedback for Non-Accepted Code* was used for non-ACCEPTED results, and *Feedback for Accepted Code (OpenHands)* for ACCEPTED results.

---

**Initial Instruction Prompt (OpenHands)**

```
You are a world-class algorithm engineer, and you are very good at programming.
    Now, you are participating in a programming contest. You are asked to solve a
    heuristic problem, known as an NP-hard problem. The duration of the contest
    is ${Duration for the problem here} hours.

There is a problem statement at the end of this message. First, please analyze the
    problem statement. Please think about the essential points of the problem
    and possible algorithms to get higher rank in the contest.

Next, please implement your solution in **C++20 (gcc 12.2.0)**. Your solution code
    should be written in the **`/workspace/solution.cpp`**. You can use external
    libraries as follows:
${EXTERNAL_LIBRARIES}

You can run your code with the following commands if you are at the `/workspace`
    directory:
```bash
# Compile
g++-12 -std=gnu++20 -O2 -DONLINE_JUDGE -DATCODER -Wall -Wextra -mtune=native -
    march=native -fconstexpr-depth=2147483647 -fconstexpr-loop-limit=2147483647 -
    fconstexpr-ops-limit=2147483647 -I/opt/ac-library -I/opt/boost/gcc/include -L
    /opt/boost/gcc/lib -o solution.out solution.cpp -lgmpxx -lgmp -I/usr/include/
    eigen3
# Run
./solution.out
```

After you implement your solution, **you must ask user (not exit) to evaluate your
    temporary solution** whether your solution is effective or not. The user
    will read your code from `/workspace/solution.cpp` file. If the user can not
    find your solution file, the user will ask you to provide your code again.
    Please make sure that your code is exist in proper way.

The user can run your solution code with 50 public cases and give you feedback
    including the score if the code is accepted or the error message if the code
    is not accepted. **Even if your solution get accepted, you must refine your
    solution or try another approach to get a better score.**

You can use the internet to search an algorithm or a library to solve the problem.
    But you must implement your solution by yourself (no plagiarism) and do not
    see any contents related to this problem. More specifically, you must not see
    web pages that contain the word "AtCoder", "AHC", "${Problem name here}" or
    contents considered to be directly related to this problem because some
    people were already solved this problem. We will check web pages you visited
    in order to detect the plagiarism and cheating. If we detect that you are
    violating this rule, you will be disqualified from the contest.

[Problem statement]
Execution time limit: ${TIME_LIMIT} sec / Memory limit: ${MEMORY_LIMIT} MB
${PROBLEM_STATEMENT}
```

---

**Feedback for Accepted Code (OpenHands)**

```
[Public test result]
Overall judge result: ACCEPTED
Overall absolute score: ${(Overall score is provided here.)}$
- Case 1: ${(Score for case 1 is provided here.)}$
- Case 2: ${(Score for case 2 is provided here.)}$
...(omission)...
```

```
- Case $N: ${(Score for case $N is provided here.)}$
You can see your solution in `http://172.17.0.1:${Visualizer port number is
    provided here.}?lang=en` to visualize your solution. If you want to visualize
     your solution in the local web UI, please access the link above. Then,
    please detect the Input field in the web page (recognize its \"bid\") and
    paste ```txt
${Input string for case 1 is provided here.}
``` to the input field. Also, please detect the Output field in the web page (
    recognize its \"bid\") and paste ```txt
${Output string for case 1 is provided here.}
``` to the output field in the page. If you do this correctly, you can see the
    visualization of your solution under the Input and Output fields in the web
    page. This visualization will help you to understand the problem and the
    features of your solution.
```

## Refinement Instruction Prompt (OpenHands)

```
${The feedback for the latest code here.}

Based on the above feedback, please consider the ways to improve your solution.
    Firstly, please analyze this given feedback and list what insights can be
    gained from it. Then, based on the insights, please refine your code to
    achieve better performance. It can be a simple bug fix, the introduction of a
     new algorithm, or any degree of change from minor to major. If you think you
    did your best on the task, please finish the interaction.
Again, you should implement your entire solution in the `/workspace/solution.cpp`
    file.
```

These OpenHands experiments used the full ALE-Bench. Three `c6i.32xlarge` instances ran in parallel, each dedicated to one of three models: GPT-4.1 mini, o4-mini-high, and Gemini 2.5 Pro. Each model processed all 40 problems, with up to four problems per instance concurrently. Although the time limit was four hours per problem, frequent agent exits before the limit allowed completion within one day.

Next, we evaluated our ALE-Agent using the lite version of ALE-Bench. Three `c6i.32xlarge` instances ran experiments for three ALE-Agent configurations in parallel: (1) Base, (2) Base + Method 1, and (3) Base + Method 1&2 (details in Appendix B). Each configuration processed all 10 problems from the lite ALE-Bench. With up to four problems concurrently per instance (subject to the 4-hour/contest duration limit), these ALE-Agent experiments took approximately half a day.

**Results.** To compare results across benchmark versions (e.g., for OpenHands which was run on the full version but compared against ALE-Agent on the lite version), we converted full version results to their lite version equivalents. This conversion reuses the private evaluation outcomes from the full version. Specifically, for each of the 10 problems included in the lite version, we extracted the results corresponding only to the test cases used in the lite version. Based on these subsetted results, we then recomputed the rank and performance score for each model on each of these 10 problems as if they were evaluated only on the lite version's test cases.

While the main paper discusses the results for ALE-Agent (on the lite version) and the lite version-converted results for OpenHands, Table 4 in this appendix presents the original full version results for OpenHands, shown alongside the iterative-refinement experiment results (which were also on the full version, initially summarized in Table 3). Furthermore, detailed per-problem performance for the ALE-Agent experiments (on the lite version) is provided in Table A6.

The ALE-Agent (Base + Method 1&2) achieved a performance score of 2880 on AHC039, corresponding to 5th place in the original contest. We submitted this solution to AtCoder for verification.[12] Analysis revealed that the solution employed Simulated Annealing, a technique LLMs also showed proficiency with in the iterative-refinement setting (cf. o4-mini-high). The official contest leaderboard[13] confirms this score matches 5th place performance in the actual AHC039, indicating ALE-Bench's scoring reproducibility. Figure A5 illustrates ALE-Agent's search tree for AHC039.

---

[12] https://atcoder.jp/contests/ahc039/submissions/65734651 (Retrieved: May 15, 2025)

[13] https://atcoder.jp/contests/ahc039/standings/extended (Retrieved: May 15, 2025) *Login required.*

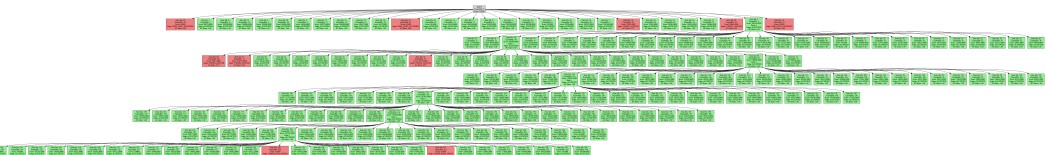

Figure A5: **The actual search tree of ALE-Agent on** `ahc039`. Each node represents one generated answering program with its public evaluation result.

Though configured to generate 30 child nodes (alternative solutions/refinements) per parent, and requests were dispatched with slight delays to manage API rates, occasional API call failures (e.g., Gemini internal errors) meant the target of 30 expanded child nodes was not always met.

## C.5    Further Analysis

### C.5.1    Statistical Significance of One-Shot Result

To assess the statistical significance, we performed five independent runs for several models under the One-Shot (C++20) setting. The average performance scores, along with their 95% confidence intervals (CIs), are presented in  Table A7a. These intervals provide a clear measure of performance stability across multiple executions. Also, for a more rigorous comparison between model pairs, we conducted a Wilcoxon signed-rank test. The results of this analysis are shown in  Table A7b. Each cell in the table contains the p-value for a one-sided test. The null hypothesis assumes no performance difference between the two models. The alternative hypothesis posits that the row model performs better than the column model. Consequently, a value below 0.05 signifies a statistically significant outperformance by the row model at the 95% confidence level. The outcomes of this test statistically substantiate several performance relationships, such as o3-high significantly outperforming o4-mini-high, GPT-4.1, and all other models. These results support that ALE-Bench produce robust and clearly differentiable performance measurements.

### C.5.2    Controlling LLM Calls in Iterative-Refinement Setting

Our primary time-based evaluation in the Iterative-Refinement setting was designed to mirror the real-world environment of competitive programming contests. However, this approach can favor models with higher inference speeds. For instance, while current LLMs are being optimized for inference efficiency, if models with different NN architectures emerge, evaluating them solely on real-time could be disadvantageous to these new models. To provide a more nuanced view that distinguishes pure model capability from the combined effects of capability and efficiency, we conducted an additional analysis. This new evaluation controls for the number of LLM calls, offering a complementary perspective. The results of this analysis are presented in Table A8. We tracked the performance scores and corresponding rank percentiles of each model after 10, 30, 50, 100, and 200 code generation attempts. The "Final" column shows the results from our original time-based setting. These final results were rerun for this analysis, which explains the slight variations from the values presented in Table 3 of the main text.

A key finding from this analysis is the difference in performance progression between DeepSeek-R1 and GPT-4.1-mini. In the early stages with fewer LLM calls, a large performance gap of nearly 200 points existed between the two models. Despite this initial disparity, their final results in the time-limited setting were very similar. This can be attributed to the substantial difference in their code generation throughput. DeepSeek-R1 generated approximately 60 codes in four hours, whereas GPT-4.1-mini produced around 300 in the same period. This observation highlights the utility of evaluating model performance by both wall time and the number of LLM calls (i.e. the number of code generations). Each metric provides a distinct and valuable perspective on a model's capabilities.

### C.5.3    Performance Differences across Genres

We found that the AI systems achieved relatively high performance on routing-type problems, where the objective is to find optimal paths. These problems are well-studied in the literature, and the systems were able to apply standard neighborhood operations such as 2-opt effectively, which likely

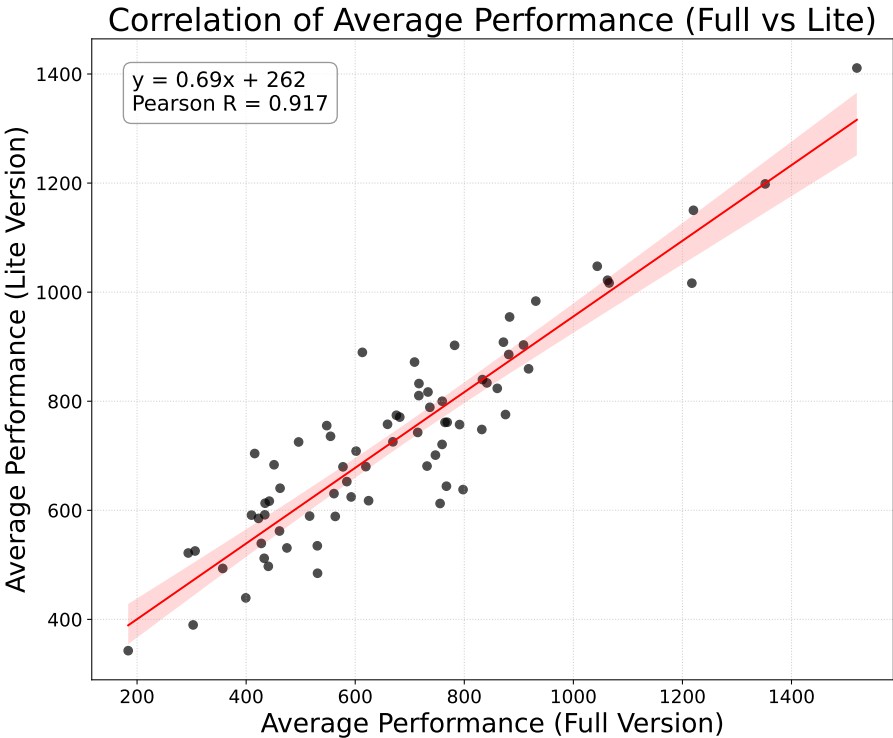

Figure A6: **Correlation of the average performance between the lite and full versions.** The red band shows the 95 % confidence interval estimated by 10000 bootstrap samples with random seed zero.

contributed to their success. In contrast, performance on planning-type problems, which involve generating sequences of actions, was lower. This appears to be due to the need for problem-specific neighborhood structures and evaluation functions, which are more difficult for the AI systems to construct automatically.

### C.5.4 Correlations between Full Version and Lite Version

We used all 73 results from the full ALE-Bench: 22 models $\times$ 3 languages (one-shot), 4 models $\times$ 1 language (C++20, iterative-refinement), and 3 models $\times$ 1 language (C++20, OpenHands scaffolding). By correlating these with their lite version-converted counterparts (see Appendix C.4), we assessed the lite version's validity for comparative AI system evaluations. Figure A6 plots average performance on the full vs. lite versions for these 73 data points. Pearson's $r = 0.9174$, Spearman's $\rho = 0.8677$, and Kendall's $\tau = 0.6963$ (all $p < 0.001$) indicate strong, statistically significant correlations. These results show a strong correlation. However, the regression slope in Figure A6 is $\approx 0.7$. This, and the lite version's design (intentionally including harder problems, see Appendix A.1), suggests the lite version is, on average, more difficult. Comparing relative AI strengths within the same version (full or lite) is valid while directly comparing across versions is not recommended as it could be misleading.

### C.5.5 Real Contest Participation

With AtCoder's permission, an in-development ALE-Agent participated in AHC046 in real time via a dedicated account (`fishylene`). The agent used only Method 2 (search tree exploration), constructing two parallel search trees, each with a target width of 50 child nodes per expansion. It checked its best solution every five minutes, submitting to AtCoder only upon improvement. The ALE-Agent placed 154th, with a performance score of 1915, as per the official contest leaderboard.[14]

---

[14]`https://atcoder.jp/contests/ahc046/standings` (Retrieved: May 15, 2025) *Login required.*

## C.6 Limitations of Experiments

We have reported on a comprehensive suite of experiments conducted on current LLMs and scaffolding with our proposed benchmark. While these investigations offer valuable insights into their respective capabilities and the utility of our benchmark, we acknowledge the following limitations that provide avenues for future research: *(1) Limited statistical robustness.* The findings presented are based on single experimental runs for each configuration. While these initial results are promising, we acknowledge that further experiments involving multiple runs are essential to establish the statistical validity of the performance claims and account for potential variability. *(2) Insufficient verification of multimodal/agent performance.* We have not used the image input function in our experiments, and further verification is needed in this area. In addition, we provide several other features, such as input case generation and visualizers, in addition to public eval/private eval. However, we have not conducted sufficient experiments to explicitly utilize these features using LLM's tool usage feature. ALE-Bench is still in the exploratory stage of development, and further investigation is needed.

Table A3: **Comparison of frontier LLMs in the one-shot setting.** We report all results in Section 5.1.

| Model | Code Language | Average Perf. | | | Perf. Distribution (%) | | | | | | | Rating | | Cost ($) | |
|---|---|---|---|---|---|---|---|---|---|---|---|---|---|---|---|
| | | short | long | overall | ≥ 400 | ≥ 800 | ≥ 1200 | ≥ 1600 | ≥ 2000 | ≥ 2400 | ≥ 2800 | raw | rank (%) | /problem | /response |
| *Non-Reasoning Models:* | | | | | | | | | | | | | | | |
| GPT-4o mini | C++20 | 442 | 420 | 433 | 50.0 | 17.5 | 2.5 | 0.0 | 0.0 | 0.0 | 0.0 | 841 | 86.7 | 0.006 | 0.002 |
| GPT-4o mini | Python3 | 401 | 494 | 441 | 52.5 | 20.0 | 0.0 | 0.0 | 0.0 | 0.0 | 0.0 | 807 | 88.2 | 0.004 | 0.001 |
| GPT-4o mini | Rust | 389 | 314 | 357 | 45.0 | 10.0 | 0.0 | 0.0 | 0.0 | 0.0 | 0.0 | 751 | 90.4 | 0.006 | 0.002 |
| GPT-4o mini | Average | 411 | 409 | 410 | 49.2 | 15.8 | 0.8 | 0.0 | 0.0 | 0.0 | 0.0 | 800 | 88.4 | 0.005 | 0.002 |
| GPT-4o | C++20 | 547 | 636 | 585 | 75.0 | 25.0 | 2.5 | 0.0 | 0.0 | 0.0 | 0.0 | 936 | 80.1 | 0.048 | 0.022 |
| GPT-4o | Python3 | 419 | 455 | 434 | 55.0 | 20.0 | 0.0 | 0.0 | 0.0 | 0.0 | 0.0 | 802 | 88.5 | 0.064 | 0.021 |
| GPT-4o | Rust | 500 | 644 | 561 | 72.5 | 30.0 | 0.0 | 0.0 | 0.0 | 0.0 | 0.0 | 887 | 83.6 | 0.066 | 0.024 |
| GPT-4o | Average | 488 | 578 | 527 | 67.5 | 25.0 | 0.8 | 0.0 | 0.0 | 0.0 | 0.0 | 875 | 84.1 | 0.059 | 0.022 |
| GPT-4.1 nano | C++20 | 443 | 489 | 462 | 60.0 | 12.5 | 0.0 | 0.0 | 0.0 | 0.0 | 0.0 | 820 | 87.5 | 0.004 | 0.001 |
| GPT-4.1 nano | Python3 | 300 | 315 | 306 | 40.0 | 5.0 | 0.0 | 0.0 | 0.0 | 0.0 | 0.0 | 657 | 94.0 | 0.004 | 0.001 |
| GPT-4.1 nano | Rust | 454 | 364 | 416 | 57.5 | 5.0 | 0.0 | 0.0 | 0.0 | 0.0 | 0.0 | 752 | 90.4 | 0.004 | 0.001 |
| GPT-4.1 nano | Average | 399 | 389 | 395 | 52.5 | 7.5 | 0.0 | 0.0 | 0.0 | 0.0 | 0.0 | 743 | 90.6 | 0.004 | 0.001 |
| GPT-4.1 mini | C++20 | 779 | 755 | 769 | 90.0 | 47.5 | 5.0 | 0.0 | 0.0 | 0.0 | 0.0 | 1135 | 67.4 | 0.009 | 0.005 |
| GPT-4.1 mini | Python3 | 634 | 732 | 676 | 85.0 | 35.0 | 0.0 | 0.0 | 0.0 | 0.0 | 0.0 | 946 | 79.7 | 0.008 | 0.005 |
| GPT-4.1 mini | Rust | 759 | 731 | 747 | 87.5 | 45.0 | 2.5 | 0.0 | 0.0 | 0.0 | 0.0 | 1047 | 73.4 | 0.010 | 0.005 |
| GPT-4.1 mini | Average | 724 | 739 | 730 | 87.5 | 42.5 | 2.5 | 0.0 | 0.0 | 0.0 | 0.0 | 1043 | 73.5 | 0.009 | 0.005 |
| GPT-4.1 | C++20 | 696 | 746 | 717 | 80.0 | 45.0 | 7.5 | 2.5 | 0.0 | 0.0 | 0.0 | 1164 | 65.1 | 0.083 | 0.035 |
| GPT-4.1 | Python3 | 831 | 753 | 798 | 87.5 | 47.5 | 12.5 | 0.0 | 0.0 | 0.0 | 0.0 | 1205 | 62.3 | 0.073 | 0.031 |
| GPT-4.1 | Rust | 734 | 728 | 732 | 80.0 | 52.5 | 10.0 | 0.0 | 0.0 | 0.0 | 0.0 | 1153 | 66.1 | 0.107 | 0.037 |
| GPT-4.1 | Average | 754 | 742 | 749 | 82.5 | 48.3 | 10.0 | 0.8 | 0.0 | 0.0 | 0.0 | 1174 | 64.5 | 0.088 | 0.034 |
| Gemini 1.5 Flash-8B | C++20 | 365 | 471 | 410 | 55.0 | 5.0 | 0.0 | 0.0 | 0.0 | 0.0 | 0.0 | 698 | 91.9 | 0.002 | 0.001 |
| Gemini 1.5 Flash-8B | Python3 | 208 | 410 | 294 | 30.0 | 15.0 | 0.0 | 0.0 | 0.0 | 0.0 | 0.0 | 787 | 89.0 | 0.001 | 0.000 |
| Gemini 1.5 Flash-8B | Rust | 129 | 258 | 184 | 25.0 | 0.0 | 0.0 | 0.0 | 0.0 | 0.0 | 0.0 | 483 | 98.2 | 0.002 | 0.000 |
| Gemini 1.5 Flash-8B | Average | 234 | 379 | 296 | 36.7 | 6.7 | 0.0 | 0.0 | 0.0 | 0.0 | 0.0 | 656 | 93.0 | 0.002 | 0.000 |
| Gemini 1.5 Flash | C++20 | 344 | 575 | 442 | 57.5 | 20.0 | 0.0 | 0.0 | 0.0 | 0.0 | 0.0 | 810 | 87.9 | 0.002 | 0.001 |
| Gemini 1.5 Flash | Python3 | 531 | 530 | 530 | 67.5 | 22.5 | 2.5 | 0.0 | 0.0 | 0.0 | 0.0 | 869 | 84.5 | 0.002 | 0.001 |
| Gemini 1.5 Flash | Rust | 260 | 361 | 303 | 35.0 | 12.5 | 0.0 | 0.0 | 0.0 | 0.0 | 0.0 | 793 | 88.8 | 0.003 | 0.001 |
| Gemini 1.5 Flash | Average | 378 | 489 | 425 | 53.3 | 18.3 | 0.8 | 0.0 | 0.0 | 0.0 | 0.0 | 824 | 87.1 | 0.002 | 0.001 |
| Gemini 1.5 Pro | C++20 | 457 | 596 | 516 | 62.5 | 30.0 | 0.0 | 0.0 | 0.0 | 0.0 | 0.0 | 892 | 83.3 | 0.044 | 0.018 |
| Gemini 1.5 Pro | Python3 | 467 | 673 | 555 | 70.0 | 27.5 | 0.0 | 0.0 | 0.0 | 0.0 | 0.0 | 927 | 80.9 | 0.020 | 0.009 |
| Gemini 1.5 Pro | Rust | 430 | 536 | 475 | 50.0 | 27.5 | 2.5 | 0.0 | 0.0 | 0.0 | 0.0 | 1017 | 75.6 | 0.033 | 0.011 |
| Gemini 1.5 Pro | Average | 451 | 602 | 515 | 60.8 | 28.3 | 0.8 | 0.0 | 0.0 | 0.0 | 0.0 | 945 | 80.0 | 0.032 | 0.013 |
| Gemini 2.0 Flash-Lite | C++20 | 466 | 432 | 451 | 55.0 | 17.5 | 5.0 | 0.0 | 0.0 | 0.0 | 0.0 | 983 | 77.7 | 0.006 | 0.002 |
| Gemini 2.0 Flash-Lite | Python3 | 423 | 451 | 435 | 55.0 | 20.0 | 0.0 | 0.0 | 0.0 | 0.0 | 0.0 | 899 | 82.8 | 0.004 | 0.001 |
| Gemini 2.0 Flash-Lite | Rust | 394 | 406 | 399 | 50.0 | 20.0 | 0.0 | 0.0 | 0.0 | 0.0 | 0.0 | 788 | 88.9 | 0.005 | 0.001 |
| Gemini 2.0 Flash-Lite | Average | 428 | 430 | 428 | 53.3 | 19.2 | 1.7 | 0.0 | 0.0 | 0.0 | 0.0 | 890 | 83.2 | 0.005 | 0.001 |
| Gemini 2.0 Flash | C++20 | 547 | 585 | 563 | 62.5 | 35.0 | 5.0 | 2.5 | 0.0 | 0.0 | 0.0 | 1031 | 74.6 | 0.006 | 0.002 |
| Gemini 2.0 Flash | Python3 | 453 | 555 | 496 | 55.0 | 30.0 | 2.5 | 0.0 | 0.0 | 0.0 | 0.0 | 1001 | 76.8 | 0.005 | 0.002 |
| Gemini 2.0 Flash | Rust | 482 | 597 | 531 | 77.5 | 15.0 | 0.0 | 0.0 | 0.0 | 0.0 | 0.0 | 840 | 86.7 | 0.005 | 0.002 |
| Gemini 2.0 Flash | Average | 494 | 579 | 530 | 65.0 | 26.7 | 2.5 | 0.8 | 0.0 | 0.0 | 0.0 | 957 | 79.4 | 0.005 | 0.002 |
| Claude 3.5 Haiku | C++20 | 464 | 457 | 461 | 65.0 | 20.0 | 0.0 | 0.0 | 0.0 | 0.0 | 0.0 | 821 | 87.4 | 0.043 | 0.013 |
| Claude 3.5 Haiku | Python3 | 620 | 619 | 620 | 77.5 | 32.5 | 5.0 | 0.0 | 0.0 | 0.0 | 0.0 | 1043 | 73.6 | 0.030 | 0.011 |
| Claude 3.5 Haiku | Rust | 439 | 400 | 423 | 52.5 | 15.0 | 2.5 | 0.0 | 0.0 | 0.0 | 0.0 | 813 | 87.7 | 0.042 | 0.013 |
| Claude 3.5 Haiku | Average | 508 | 492 | 501 | 65.0 | 22.5 | 2.5 | 0.0 | 0.0 | 0.0 | 0.0 | 892 | 82.9 | 0.038 | 0.012 |
| Claude 3.5 Sonnet | C++20 | 744 | 674 | 715 | 85.0 | 40.0 | 7.5 | 0.0 | 0.0 | 0.0 | 0.0 | 1092 | 70.5 | 0.123 | 0.055 |
| Claude 3.5 Sonnet | Python3 | 782 | 676 | 737 | 82.5 | 45.0 | 5.0 | 0.0 | 0.0 | 0.0 | 0.0 | 1114 | 68.9 | 0.099 | 0.039 |
| Claude 3.5 Sonnet | Rust | 633 | 538 | 593 | 72.5 | 30.0 | 5.0 | 0.0 | 0.0 | 0.0 | 0.0 | 993 | 77.1 | 0.126 | 0.040 |
| Claude 3.5 Sonnet | Average | 720 | 629 | 681 | 80.0 | 38.3 | 5.8 | 0.0 | 0.0 | 0.0 | 0.0 | 1066 | 72.2 | 0.116 | 0.045 |
| Claude 3.7 Sonnet | C++20 | 851 | 810 | 833 | 90.0 | 55.0 | 17.5 | 0.0 | 0.0 | 0.0 | 0.0 | 1197 | 63.2 | 0.287 | 0.142 |
| Claude 3.7 Sonnet | Python3 | 762 | 809 | 782 | 82.5 | 52.5 | 7.5 | 0.0 | 0.0 | 0.0 | 0.0 | 1144 | 66.8 | 0.268 | 0.126 |
| Claude 3.7 Sonnet | Rust | 818 | 698 | 767 | 80.0 | 47.5 | 15.0 | 5.0 | 0.0 | 0.0 | 0.0 | 1212 | 61.8 | 0.427 | 0.157 |
| Claude 3.7 Sonnet | Average | 810 | 772 | 794 | 84.2 | 51.7 | 13.3 | 1.7 | 0.0 | 0.0 | 0.0 | 1184 | 63.9 | 0.328 | 0.142 |
| DeepSeek-V3 | C++20 | 638 | 688 | 659 | 75.0 | 42.5 | 10.0 | 0.0 | 0.0 | 0.0 | 0.0 | 1142 | 66.8 | 0.008 | 0.003 |
| DeepSeek-V3 | Python3 | 440 | 412 | 428 | 50.0 | 20.0 | 0.0 | 0.0 | 0.0 | 0.0 | 0.0 | 930 | 80.5 | 0.010 | 0.004 |
| DeepSeek-V3 | Rust | 617 | 581 | 602 | 70.0 | 32.5 | 2.5 | 0.0 | 0.0 | 0.0 | 0.0 | 1024 | 75.1 | 0.010 | 0.004 |
| DeepSeek-V3 | Average | 565 | 560 | 563 | 65.0 | 31.7 | 4.2 | 0.0 | 0.0 | 0.0 | 0.0 | 1032 | 74.1 | 0.009 | 0.004 |
| *Reasoning Models:* | | | | | | | | | | | | | | | |
| o1-high | C++20 | 768 | 823 | 791 | 90.0 | 52.5 | 5.0 | 0.0 | 0.0 | 0.0 | 0.0 | 1143 | 66.8 | 0.579 | 0.464 |
| o1-high | Python3 | 531 | 641 | 578 | 75.0 | 32.5 | 0.0 | 0.0 | 0.0 | 0.0 | 0.0 | 938 | 80.0 | 0.945 | 0.420 |
| o1-high | Rust | 789 | 732 | 765 | 87.5 | 50.0 | 10.0 | 0.0 | 0.0 | 0.0 | 0.0 | 1137 | 67.2 | 0.698 | 0.451 |
| o1-high | Average | 696 | 732 | 711 | 84.2 | 45.0 | 5.0 | 0.0 | 0.0 | 0.0 | 0.0 | 1073 | 71.3 | 0.741 | 0.445 |
| o3-mini-high | C++20 | 615 | 742 | 669 | 90.0 | 35.0 | 0.0 | 0.0 | 0.0 | 0.0 | 0.0 | 906 | 82.4 | 0.041 | 0.037 |
| o3-mini-high | Python3 | 820 | 677 | 759 | 95.0 | 42.5 | 5.0 | 0.0 | 0.0 | 0.0 | 0.0 | 1091 | 70.7 | 0.061 | 0.042 |
| o3-mini-high | Rust | 726 | 743 | 733 | 95.0 | 35.0 | 7.5 | 0.0 | 0.0 | 0.0 | 0.0 | 1052 | 73.0 | 0.044 | 0.039 |
| o3-mini-high | Average | 720 | 721 | 721 | 93.3 | 37.5 | 4.2 | 0.0 | 0.0 | 0.0 | 0.0 | 1016 | 75.4 | 0.049 | 0.039 |
| o3-high | C++20 | 1116 | 946 | 1044 | 97.5 | 82.5 | 22.5 | 5.0 | 0.0 | 0.0 | 0.0 | 1456 | 43.2 | 0.734 | 0.506 |
| o3-high | Python3 | 1135 | 971 | 1066 | 100.0 | 80.0 | 27.5 | 2.5 | 0.0 | 0.0 | 0.0 | 1424 | 45.4 | 0.677 | 0.423 |
| o3-high | Rust | 1119 | 987 | 1063 | 97.5 | 80.0 | 27.5 | 7.5 | 0.0 | 0.0 | 0.0 | 1532 | 38.0 | 0.801 | 0.501 |
| o3-high | Average | 1123 | 968 | 1057 | 98.3 | 80.8 | 25.8 | 5.0 | 0.0 | 0.0 | 0.0 | 1471 | 42.2 | 0.738 | 0.477 |
| o4-mini-high | C++20 | 866 | 808 | 841 | 92.5 | 60.0 | 7.5 | 0.0 | 0.0 | 0.0 | 0.0 | 1194 | 63.6 | 0.041 | 0.037 |
| o4-mini-high | Python3 | 866 | 889 | 876 | 97.5 | 67.5 | 7.5 | 0.0 | 0.0 | 0.0 | 0.0 | 1120 | 68.6 | 0.044 | 0.034 |
| o4-mini-high | Rust | 981 | 747 | 882 | 97.5 | 67.5 | 10.0 | 0.0 | 0.0 | 0.0 | 0.0 | 1215 | 61.5 | 0.048 | 0.039 |
| o4-mini-high | Average | 904 | 815 | 866 | 95.8 | 65.0 | 8.3 | 0.0 | 0.0 | 0.0 | 0.0 | 1176 | 64.6 | 0.045 | 0.037 |
| Gemini 2.5 Flash | C++20 | 905 | 827 | 872 | 82.5 | 62.5 | 20.0 | 5.0 | 0.0 | 0.0 | 0.0 | 1422 | 45.5 | 0.194 | 0.104 |
| Gemini 2.5 Flash | Python3 | 589 | 646 | 613 | 65.0 | 47.5 | 5.0 | 0.0 | 0.0 | 0.0 | 0.0 | 1157 | 65.8 | 0.283 | 0.098 |
| Gemini 2.5 Flash | Rust | 639 | 424 | 548 | 55.0 | 30.0 | 10.0 | 2.5 | 0.0 | 0.0 | 0.0 | 1237 | 59.9 | 0.316 | 0.092 |
| Gemini 2.5 Flash | Average | 711 | 632 | 678 | 67.5 | 46.7 | 11.7 | 2.5 | 0.0 | 0.0 | 0.0 | 1272 | 57.1 | 0.264 | 0.098 |
| Gemini 2.5 Pro | C++20 | 938 | 688 | 832 | 82.5 | 60.0 | 25.0 | 5.0 | 0.0 | 0.0 | 0.0 | 1373 | 49.3 | 0.472 | 0.242 |
| Gemini 2.5 Pro | Python3 | 620 | 847 | 717 | 70.0 | 52.5 | 20.0 | 0.0 | 0.0 | 0.0 | 0.0 | 1285 | 55.8 | 0.572 | 0.183 |
| Gemini 2.5 Pro | Rust | 857 | 618 | 756 | 70.0 | 57.5 | 20.0 | 5.0 | 0.0 | 0.0 | 0.0 | 1293 | 54.9 | 0.527 | 0.188 |
| Gemini 2.5 Pro | Average | 805 | 718 | 768 | 74.2 | 56.7 | 21.7 | 3.3 | 0.0 | 0.0 | 0.0 | 1317 | 53.3 | 0.524 | 0.204 |
| Claude 3.7 Sonnet (Thinking) | C++20 | 911 | 792 | 860 | 90.0 | 57.5 | 17.5 | 2.5 | 0.0 | 0.0 | 0.0 | 1328 | 52.3 | 0.375 | 0.170 |
| Claude 3.7 Sonnet (Thinking) | Python3 | 884 | 881 | 883 | 92.5 | 70.0 | 7.5 | 0.0 | 0.0 | 0.0 | 0.0 | 1198 | 63.1 | 0.314 | 0.167 |
| Claude 3.7 Sonnet (Thinking) | Rust | 972 | 876 | 931 | 97.5 | 65.0 | 15.0 | 5.0 | 0.0 | 0.0 | 0.0 | 1345 | 51.0 | 0.367 | 0.181 |
| Claude 3.7 Sonnet (Thinking) | Average | 923 | 850 | 892 | 93.3 | 64.2 | 13.3 | 2.5 | 0.0 | 0.0 | 0.0 | 1290 | 55.4 | 0.352 | 0.173 |
| DeepSeek-R1 | C++20 | 713 | 822 | 760 | 75.0 | 47.5 | 12.5 | 0.0 | 0.0 | 0.0 | 0.0 | 1206 | 62.3 | 0.063 | 0.028 |
| DeepSeek-R1 | Python3 | 711 | 706 | 709 | 75.0 | 47.5 | 10.0 | 0.0 | 0.0 | 0.0 | 0.0 | 1184 | 64.0 | 0.062 | 0.025 |
| DeepSeek-R1 | Rust | 786 | 541 | 682 | 72.5 | 47.5 | 15.0 | 0.0 | 0.0 | 0.0 | 0.0 | 1238 | 59.8 | 0.056 | 0.022 |
| DeepSeek-R1 | Average | 736 | 690 | 717 | 74.2 | 47.5 | 12.5 | 0.0 | 0.0 | 0.0 | 0.0 | 1209 | 62.0 | 0.060 | 0.025 |

Table A4: **Statistics for the actual AtCoder users who participated in contests 5 or more times.** Standard deviations in this table are calculated with using the number of users as the denominator (i.e., ddof = 0).

| Rating | #Users | #Attendance | | Average Perf. | | Perf. Distribution (%) | | | |
|---|---|---|---|---|---|---|---|---|---|
| | | mean | SD ($\sigma$) | mean | SD ($\sigma$) | $\geq 400$ | $\geq 1600$ | $\geq 2000$ | $\geq 2400$ |
| [400, 500) | 24 | 5.67 | 1.03 | 614 | 127 | 73.0 | 0.0 | 0.0 | 0.0 |
| [450, 550) | 32 | 5.72 | 1.40 | 655 | 101 | 81.3 | 0.0 | 0.0 | 0.0 |
| [500, 600) | 52 | 6.12 | 1.83 | 688 | 113 | 83.4 | 0.0 | 0.0 | 0.0 |
| [550, 650) | 62 | 6.29 | 2.03 | 725 | 121 | 85.5 | 0.0 | 0.0 | 0.0 |
| [600, 700) | 80 | 6.86 | 2.55 | 752 | 118 | 89.8 | 0.0 | 0.0 | 0.0 |
| [650, 750) | 81 | 7.35 | 2.73 | 760 | 127 | 89.6 | 0.0 | 0.0 | 0.0 |
| [700, 800) | 75 | 7.48 | 2.62 | 816 | 132 | 90.3 | 0.3 | 0.0 | 0.0 |
| [750, 850) | 100 | 7.86 | 3.34 | 863 | 125 | 93.3 | 0.4 | 0.0 | 0.0 |
| [800, 900) | 126 | 8.11 | 3.24 | 897 | 116 | 95.1 | 0.3 | 0.0 | 0.0 |
| [850, 950) | 144 | 8.44 | 3.79 | 933 | 128 | 95.0 | 0.6 | 0.0 | 0.0 |
| [900, 1000) | 133 | 8.86 | 4.37 | 967 | 137 | 94.9 | 1.6 | 0.0 | 0.0 |
| [950, 1050) | 139 | 8.51 | 3.64 | 1013 | 121 | 96.4 | 2.5 | 0.0 | 0.0 |
| [1000, 1100) | 151 | 9.26 | 4.96 | 1045 | 133 | 96.8 | 4.0 | 0.0 | 0.0 |
| [1050, 1150) | 149 | 10.63 | 5.97 | 1066 | 145 | 96.5 | 4.4 | 0.0 | 0.0 |
| [1100, 1200) | 156 | 10.59 | 5.56 | 1099 | 143 | 96.8 | 5.7 | 0.2 | 0.0 |
| [1150, 1250) | 171 | 10.88 | 5.54 | 1135 | 149 | 97.0 | 8.8 | 0.4 | 0.0 |
| [1200, 1300) | 188 | 11.34 | 5.64 | 1168 | 147 | 97.5 | 10.6 | 0.4 | 0.0 |
| [1250, 1350) | 177 | 11.45 | 5.95 | 1210 | 146 | 97.7 | 13.5 | 0.6 | 0.0 |
| [1300, 1400) | 165 | 12.29 | 6.35 | 1238 | 150 | 97.6 | 16.3 | 1.4 | 0.0 |
| [1350, 1450) | 157 | 12.80 | 6.56 | 1266 | 167 | 98.0 | 18.7 | 2.2 | 0.0 |
| [1400, 1500) | 154 | 13.75 | 7.44 | 1287 | 169 | 97.8 | 20.8 | 1.6 | 0.0 |
| [1450, 1550) | 159 | 15.29 | 8.07 | 1301 | 166 | 97.2 | 22.3 | 2.8 | 0.0 |
| [1500, 1600) | 162 | 16.27 | 8.48 | 1348 | 160 | 97.7 | 26.3 | 4.5 | 0.0 |
| [1550, 1650) | 155 | 16.60 | 8.77 | 1380 | 159 | 98.1 | 30.5 | 4.7 | 0.2 |
| [1600, 1700) | 136 | 16.27 | 8.45 | 1408 | 158 | 97.9 | 33.9 | 6.5 | 0.2 |
| [1650, 1750) | 121 | 17.26 | 9.21 | 1434 | 145 | 98.0 | 35.2 | 7.5 | 0.6 |
| [1700, 1800) | 102 | 19.07 | 10.17 | 1458 | 174 | 98.3 | 38.6 | 8.1 | 1.2 |
| [1750, 1850) | 102 | 20.57 | 9.04 | 1479 | 169 | 98.8 | 41.0 | 9.4 | 0.8 |
| [1800, 1900) | 87 | 21.92 | 8.41 | 1487 | 152 | 98.7 | 41.5 | 10.3 | 0.8 |
| [1850, 1950) | 83 | 21.71 | 9.55 | 1550 | 175 | 98.3 | 46.8 | 14.7 | 2.0 |
| [1900, 2000) | 78 | 23.33 | 10.57 | 1578 | 167 | 98.1 | 49.4 | 16.7 | 2.2 |
| [1950, 2050) | 60 | 25.47 | 9.61 | 1606 | 126 | 98.6 | 52.1 | 18.6 | 2.1 |
| [2000, 2100) | 64 | 24.31 | 10.14 | 1643 | 130 | 99.0 | 55.2 | 21.5 | 3.2 |
| [2050, 2150) | 56 | 26.88 | 11.19 | 1653 | 135 | 98.8 | 56.2 | 22.5 | 4.0 |
| [2100, 2200) | 58 | 27.17 | 10.70 | 1729 | 182 | 98.6 | 63.2 | 30.2 | 5.8 |
| [2150, 2250) | 57 | 27.09 | 12.77 | 1782 | 201 | 98.3 | 67.5 | 36.6 | 8.3 |
| [2200, 2300) | 47 | 28.53 | 12.94 | 1786 | 181 | 98.4 | 66.9 | 36.5 | 9.1 |
| [2250, 2350) | 47 | 31.51 | 11.50 | 1808 | 148 | 98.6 | 70.1 | 37.4 | 9.1 |
| [2300, 2400) | 45 | 32.02 | 11.67 | 1854 | 157 | 99.0 | 72.0 | 40.4 | 12.0 |
| [2350, 2450) | 34 | 28.35 | 11.27 | 1947 | 202 | 99.7 | 76.4 | 45.8 | 17.1 |
| [2400, 2500) | 30 | 28.10 | 10.62 | 1993 | 232 | 99.7 | 79.3 | 48.6 | 20.9 |
| [2450, 2550) | 30 | 27.67 | 10.45 | 2030 | 180 | 99.3 | 80.5 | 53.7 | 24.5 |
| [2500, 2600) | 22 | 31.32 | 11.18 | 2066 | 127 | 99.2 | 82.7 | 59.7 | 27.0 |

Table A5: **The full version result with scaffolding.** The experiment of ALE-Agent with the full version setting was not conducted.

| Scaffolding | Average Perf. | | | Perf. Distribution (%) | | | | | | | Rating | | Cost ($) | |
|---|---|---|---|---|---|---|---|---|---|---|---|---|---|---|
| | short | long | overall | rank (%) | $\geq 400$ | $\geq 800$ | $\geq 1200$ | $\geq 1600$ | $\geq 2000$ | $\geq 2400$ | $\geq 2800$ | raw | rank (%) | /problem | /response |
| *Sequential Refinement:* | | | | | | | | | | | | | | | |
| GPT-4.1 mini | 1293 | 1114 | 1217 | 51.5 | 100.0 | 95.0 | 42.5 | 17.5 | 2.5 | 0.0 | 0.0 | 1636 | 30.5 | 2.137 | 0.010 |
| o4-mini-high | 1677 | 1307 | 1520 | 22.3 | 100.0 | 97.5 | 87.5 | 32.5 | 15.0 | 5.0 | 0.0 | 2104 | 11.8 | 7.174 | 0.047 |
| Gemini 2.5 Pro | 1389 | 1301 | 1352 | 36.8 | 95.0 | 92.5 | 62.5 | 27.5 | 7.5 | 5.0 | 0.0 | 1960 | 15.7 | 11.126 | 0.134 |
| DeepSeek-R1 | 1268 | 1155 | 1220 | 51.1 | 97.5 | 87.5 | 50.0 | 15.0 | 5.0 | 2.5 | 0.0 | 1891 | 18.3 | 1.141 | 0.024 |
| *OpenHands [35]:* | | | | | | | | | | | | | | | |
| GPT-4.1 mini | 687 | 540 | 625 | 96.8 | 80.0 | 32.5 | 2.5 | 0.0 | 0.0 | 0.0 | 0.0 | 996 | 77.0 | 0.114 | 0.005 |
| o4-mini-high | 991 | 818 | 918 | 81.6 | 85.0 | 72.5 | 22.5 | 2.5 | 2.5 | 0.0 | 0.0 | 1469 | 42.2 | 2.321 | 0.053 |
| Gemini 2.5 Pro | 923 | 890 | 909 | 82.4 | 82.5 | 67.5 | 30.0 | 2.5 | 0.0 | 0.0 | 0.0 | 1386 | 48.2 | 3.331 | 0.149 |

Table A6: **Performance for each problem.** We report the performance of each problem from results in Section 5.2 and Section 5.3. Rows with gray background correspond to long contests, while white rows correspond to short contests.

| Problem ID | Sequential Refinement | | | | ALE-Agent | | |
|---|---|---|---|---|---|---|---|
| | GPT-4.1 mini | o4-mini-high | Gemini 2.5 Pro | DeepSeek-R1 | Base | + Method 1 | + Method 1&2 |
| ahc001 | 1016 | 1288 | 1760 | 383 | – | – | – |
| ahc002 | 1168 | 1238 | 1205 | 842 | – | – | – |
| ahc003 | 1769 | 1602 | 1885 | 1632 | – | – | – |
| ahc004 | 1723 | 1726 | 1155 | 1560 | – | – | – |
| ahc005 | 1665 | 2107 | 1673 | 1582 | – | – | – |
| future-contest-2022-qual | 1518 | 1552 | 1702 | 1530 | – | – | – |
| ahc006 | 1322 | 2472 | 2456 | 2440 | – | – | – |
| ahc007 | 1021 | 1249 | 1906 | 854 | – | – | – |
| ahc008 | 1151 | 1217 | 1182 | 1006 | 1075 | 1061 | 1189 |
| ahc009 | 1077 | 791 | 2446 | 841 | – | – | – |
| ahc010 | 1167 | 1565 | 1441 | 1099 | – | – | – |
| ahc011 | 1148 | 1562 | 1403 | 1221 | 1447 | 1531 | 1652 |
| ahc012 | 977 | 2236 | 123 | 674 | – | – | – |
| ahc014 | 1016 | 1053 | 999 | 1254 | – | – | – |
| ahc015 | 1237 | 1317 | 1415 | 1142 | 1265 | 1315 | 2446 |
| ahc016 | 506 | 1515 | 966 | 1514 | 1262 | 1199 | 1457 |
| ahc017 | 1241 | 1224 | 1190 | 1260 | – | – | – |
| ahc019 | 893 | 1256 | 1016 | 1009 | – | – | – |
| ahc020 | 1395 | 2545 | -70 | 1726 | – | – | – |
| ahc021 | 1633 | 1319 | 1619 | 1481 | – | – | – |
| toyota2023summer-final | 1018 | 1662 | 986 | 1062 | – | – | – |
| ahc024 | 717 | 1283 | 1517 | 1320 | 1243 | 1830 | 1980 |
| ahc025 | 1210 | 1209 | 1216 | 1210 | 1113 | 886 | 1331 |
| ahc026 | 1003 | 1402 | 1313 | 712 | 712 | 1320 | 1965 |
| ahc027 | 1074 | 1358 | 1374 | 557 | 1168 | 719 | 1740 |
| ahc028 | 1868 | 1986 | 1912 | 1868 | – | – | – |
| ahc030 | 1037 | 1017 | 1038 | 1356 | – | – | – |
| ahc031 | 979 | 1333 | 1549 | 1299 | – | – | – |
| ahc032 | 1202 | 1361 | 1432 | 936 | – | – | – |
| ahc033 | 959 | 959 | 971 | 848 | – | – | – |
| ahc034 | 1769 | 1793 | 1020 | 915 | – | – | – |
| ahc035 | 1257 | 1638 | 1592 | 2348 | – | – | – |
| ahc038 | 1038 | 1244 | 1218 | 919 | – | – | – |
| ahc039 | 983 | 1686 | 817 | 1658 | 1661 | 2039 | 2880 |
| ahc040 | 1113 | 1383 | 1187 | 1477 | – | – | – |
| ahc041 | 2049 | 2306 | 2372 | 1203 | – | – | – |
| ahc042 | 1255 | 1181 | 1253 | 594 | – | – | – |
| ahc044 | 1124 | 2150 | 1628 | 1194 | – | – | – |
| ahc045 | 1275 | 1452 | 1469 | 1162 | – | – | – |
| ahc046 | 1119 | 1558 | 737 | 1119 | 725 | 737 | 2153 |

Table A7: **Statistical significance analysis of One-Shot setting.** Table (a) shows the average scores with 95% confidence intervals over five runs. Table (b) presents p-values from a one-sided Wilcoxon signed-rank test, corresponding to the null hypothesis of no performance difference between the row and column models.

(a) Average Performance (95% CI).

| Model | Performance |
|---|---|
| GPT-4.1 mini | $732.29 \pm 27.90$ |
| GPT-4.1 | $780.81 \pm 81.23$ |
| o3-high | $1057.81 \pm 46.74$ |
| o4-mini-high | $882.13 \pm 30.29$ |
| Gemini 2.0 Flash | $565.29 \pm 55.97$ |
| Claude 3.7 Sonnet (Thinking) | $858.06 \pm 17.95$ |
| DeepSeek-R1 | $854.65 \pm 40.72$ |

(b) P-values from Wilcoxon signed-rank test.

| Model | GPT-4.1 mini | GPT-4.1 | o3-high | o4-mini-high | Gemini 2.0 Flash | Claude 3.7 Sonnet (Thinking) | DeepSeek-R1 |
|---|---|---|---|---|---|---|---|
| GPT-4.1 mini | – | 0.906 | 1.000 | 1.000 | 0.031 | 1.000 | 1.000 |
| GPT-4.1 | 0.156 | – | 1.000 | 1.000 | 0.031 | 1.000 | 1.000 |
| o3-high | 0.031 | 0.031 | – | 0.031 | 0.031 | 0.031 | 0.031 |
| o4-mini-high | 0.031 | 0.031 | 1.000 | – | 0.031 | 0.156 | 0.156 |
| Gemini 2.0 Flash | 1.000 | 1.000 | 1.000 | 1.000 | – | 1.000 | 1.000 |
| Claude 3.7 Sonnet (Thinking) | 0.031 | 0.031 | 1.000 | 0.906 | 0.031 | – | 0.594 |
| DeepSeek-R1 | 0.031 | 0.031 | 1.000 | 0.906 | 0.031 | 0.500 | – |

Table A8: **Iterative-Refinement result with fixed number of LLM calls.** Rank percentiles are shown in parentheses. The "Final" column corresponds to the original time-limited setting.

| Model | 10 | 30 | 50 | 100 | 200 | Final |
|---|---|---|---|---|---|---|
| GPT-4.1 mini | 924 (81.08%) | 1010 (74.01%) | 1028 (71.94%) | 1109 (63.33%) | 1205 (53.06%) | 1218 (51.49%) |
| o4-mini-high | 1262 (46.89%) | 1364 (35.59%) | 1419 (30.63%) | 1494 (24.37%) | 1522 (22.25%) | 1522 (22.25%) |
| Gemini 2.5 Pro | 1212 (52.12%) | 1240 (49.05%) | 1314 (41.53%) | 1348 (37.48%) | 1351 (36.76%) | 1351 (36.76%) |
| DeepSeek-R1 | 1107 (63.69%) | 1153 (58.06%) | 1215 (51.89%) | 1221 (51.13%) | 1221 (51.13%) | 1221 (51.13%) |

