# OpenReview forum: "ALE-Bench: A Benchmark for Long-Horizon Objective-Driven Algorithm Engineering"
_NeurIPS.cc/2025/Datasets_and_Benchmarks_Track — NeurIPS 2025 Datasets and Benchmarks Track poster_

### Official Review · Reviewer_B98W · 2025-07-01

**Rating:** 5
**Confidence:** 4

**Summary:**

ALE-Bench is a new benchmark for long-horizon algorithm engineering, built upon real-world optimization problems from the AtCoder Heuristic Contest (AHC). It supports interactive code refinement, visualization, and standardized scoring to evaluate how well AI systems can iteratively improve solutions to complex, score-based programming tasks with no known optimal solution. The paper also introduces ALE-Agent, an agent framework that combines LLMs with search strategies to enhance performance.

**Additional Feedback:**

1. Suggest expanding problem diversity and synthetic variations to avoid overfitting.

2. Explore integration of multi-agent frameworks or reinforcement learning agents.

**Dataset Code Accessibility:**

Yes

**Dataset Code Comments:**

Dataset: https://huggingface.co/datasets/SakanaAI/ALE-Bench

Code: https://github.com/SakanaAI/ALE-Bench

Includes Dockerfiles, documentation, and AWS deployment scripts for full reproducibility.

**Ethical Comments:**

The dataset contains no personal or sensitive information. Participation in AHC is voluntary and public, and the authors explicitly discourage covert AI participation in live contests.

**Ethical Considerations:**

No, there are no or only very minor ethics concerns

**Final Justification:**

My concern has been addressed, and since my initial score was already quite high, I will maintain my original rating.

**Limitations Weaknesses:**

1. Limited Dataset Size: Only 40 contests included; future versions should expand to maintain challenge diversity.

2. Potential Training Data Contamination: Since AHC problems are public, they may have been included in LLM training data.

3. Language Performance Discrepancy: C++ consistently outperforms Python and Rust, raising questions about cross-language fairness.

**Strengths Contributions:**

- Novelty and Significance: Fills a gap in AI benchmarking by focusing on long-term reasoning and continuous optimization, going beyond short-duration coding benchmarks.

- Realism and Relevance: Uses real contest data from AtCoder, aligning closely with industrial applications such as logistics and scheduling.

- Comprehensive Evaluation Framework: Supports multiple languages (C++, Python, Rust), visualizations, and detailed metrics like performance and rating.

- Agent Design Innovation: Introduces ALE-Agent, which combines LLMs with search and prompting strategies to significantly boost model performance.

- Openness and Reproducibility: Data and code are publicly available. Docker and AWS scripts ensure reproducibility and fair comparison with human contestants.

---

> ### Author Rebuttal · Authors · 2025-07-30
>
> First, we would like to express our deep gratitude for the reviewer's understanding of the significance of this research and for the positive evaluation.
>
> > Fills a gap in AI benchmarking by focusing on long-term reasoning and continuous optimization, going beyond short-duration coding benchmarks.
>
> This is precisely our view, and we believe it is crucial in filling a gap in AI benchmarking.
>
> > Docker and AWS scripts ensure reproducibility and fair comparison with human contestants.
>
> This is a feature we developed with great care, and we are extremely honored that the reviewer has highly praised it.
>
> Regarding the points the reviewer raised as "Limitations Weaknesses," we would like to state our opinions below.
> We have:
>
> - Addressed the concerns regarding dataset size and data contamination by outlining our plans for expansion and continuous monitoring (**[W1, W2]**).
> - Provided a deeper analysis of the language performance discrepancy, framing it as an interesting insight into both real-world constraints and LLM capabilities (**[W3]**).
>
> Although the rebuttal format precludes the submission of a revised manuscript, we have incorporated all the revisions based on the reviewer's suggestion.
>
> Furthermore, regarding the "Additional Feedback" the reviewer proposed to further develop this research, we discussed the future research directions opened by ALE-Bench, including the potential for synthetic problem generation and advanced AI agents.
> We feel it is important to include these as "Future Work" in Chapter 6 or the Appendix as needed.
>
> ---
>
> ### [W1] Limited Dataset Size
>
> We acknowledge that the benchmark size of 40 is relatively small compared to other benchmarks.
> We plan to continue adding newly held contests to the dataset in the future and we have already reached an agreement with the AHC organizer to incorporate newly released contests.
> The benchmark framework is also designed to make such expansions straightforward.
>
> However, as detailed in the limitations (Section 6), it is already expected that the variance of the results is reduced due to the nature of each problem requiring prolonged trial-and-error.
> Furthermore, this number of problems is considered a reliable indicator of skill among human competitors.
> Since a single problem is worked on for a long time, the data obtained is very dense, and we believe it contains sufficient information to evaluate an AI system's capabilities from multiple perspectives.
>
> ---
>
> ### [W2] Contamination Risks
>
> As the reviewer pointed out, since AHC is publicly accessible on the web, there is a possibility that it has been included in the training corpora of LLMs.
> However, as discussed in Section 5.4, we have not observed severe contamination effects so far.
>
> We have also, with special permission from AtCoder, had an AI agent participate in an actual AHC and have reported the outcomes (Appendix C.5.2).
> While contamination risks may arise in the future, we plan to establish a system for continuous monitoring, in conjunction with our future plans for adding new contests related to W1.
>
> ---
>
> ### [W3] Language Performance Discrepancy
>
> It is true that there is indeed a performance gap between languages, but at the same time, it provides us with interesting insights.
>
> As mentioned in Section 3.2, ALE-Bench is primarily designed to replicate the contest environment itself for AI.
> Therefore, the rules regarding the specifications of each programming language are identical to those faced by human participants, and we are not aiming to make the languages fair to each other.
> Because AHC tasks are typically CPU-bound, native compiled languages (C++ and Rust) are very fast and have an advantage over Python.
> There is generally no inherent advantage or disadvantage between C++ and Rust, and indeed, looking at the submissions of top contestants, both C++ and Rust are used.
> This situation mirrors real-world combinatorial optimization practice, where execution speed is a crucial factor.
> Interestingly, on the other hand, Table 2 shows a gap even between Rust and C++.
> This suggests that our benchmark may have highlighted differences beyond mere execution speed, such as the LLM's fluency or the quality of the generated code.
>
> We have expanded on this interesting discussion in the Appendix.
> Thank the reviewer for prompting this interesting analysis.
>
> ---
>
> ### [A1] Synthetic Problems
>
> From the perspective of synthetic variations of problems, it may be possible to avoid overfitting through augmentation, such as changing the story or notation of the problem statements by using LLMs.
> Also, beyond simple transformations, one could consider initiatives to generate completely new problems including visualization tools based on the real contest data, although this would make an apple-to-apple comparison with humans impossible.
> The mere act of generating synthetic problems encompasses a wide range of variations and required levels of sophistication, from small perturbations to wholly novel problems that demand human-level creativity, constituting a research topic of its own.
>
> ---
>
> ### [A2] More Exploration of Agents
>
> We are aware that research on AI agents is a highly active area, and we hope that ALE-Bench will help to stimulate further research in the community.
>
> For exploring more advanced AI agents, we believe that multi-agent setups (e.g., specialized sub-agents for analysis, implementation, ideation) or systems that leverage multiple LLMs (o3, Gemini 2.5, Claude 4, etc.) are straightforward to integrate with ALE-Bench.
>
> Also, related to A1, if larger problem pools can be created by LLMs, further research such as training reinforcement learning agents to improve open-source models would also become possible.
>
> ---
>
> All the points in the review were constructive and forward-looking, and we are deeply grateful for the opportunity to strengthen the paper accordingly.

---

### Official Review · Reviewer_1rad · 2025-07-02

**Rating:** 5
**Confidence:** 3

**Summary:**

The work introduces ALE-Bench, a benchmark designed to evaluate the performance of AI systems in algorithm engineering across complex, long-horizon optimization tasks. By utilizing real problems from the AtCoder Heuristic Contest (AHC), ALE-Bench emphasizes iterative solution refinement rather than binary success or failure. The authors highlight the performance gap between leading large language models (LLMs) and human experts, indicating the need for advanced reasoning capabilities in AI.

**Additional Feedback:**

**Technical Quality**: The paper presents a solid technical framework with ALE-Bench, offering an innovative benchmark for evaluating AI systems in algorithm engineering.

**Impact**: This work has potential high impact in evaluating long-horizon optimization tasks, particularly in revealing gaps between AI systems and human experts.

**Evaluation and Resources**: While the evaluation framework is good, the limited dataset size and potential contamination issues constrain the generalizability of the results.

**Ethical Considerations**: There are no significant ethical issues, but a deeper discussion on dataset usage and potential risks would enhance the paper.

**Dataset Code Accessibility:**

Yes

**Ethical Considerations:**

No, there are no or only very minor ethics concerns

**Final Justification:**

I have read the author rebuttal carefully and participated in the reviewer discussion. The authors have adequately addressed all four of my concerns in the rebuttal. I appreciate their clarifications and the additional analyses. I have no further concern at this stage.

1. This work introduces a novel and realistic benchmark focused on long-horizon, score-based optimization tasks, addressing a significant gap in existing AI evaluation frameworks.
2. Its use of real-world problems, detailed metrics, and support for interactive solution strategies makes it a valuable contribution to advancing the evaluation of AI systems in complex algorithmic settings.

In summary, I believe it meets the standard for acceptance.

**Limitations Weaknesses:**

**1. Limited Dataset Size**: The benchmark is based on a relatively small number of problems (40), which may restrict the generalizability of the results. The authors acknowledge this limitation, suggesting that future work should focus on expanding the dataset.

**2. Potential Contamination Risks**: As the problems are publicly available, there is a risk of contamination in AI training data, which could affect the validity of the benchmark results. Although the authors report no significant contamination effects, continuous monitoring is necessary.

**3. Performance Variability**: The paper does not extensively discuss the variability of AI performance across different problem types, which may affect the reliability of the benchmark as a universal evaluation tool.

**4. Lack of Extensive Comparison**: While the authors compare AI systems against human experts, there is limited exploration of how different AI approaches can be optimized within the context of ALE-Bench.

**Strengths Contributions:**

**1. Novel Benchmarking Framework**: ALE-Bench fills a gap in the current landscape of AI benchmarks by focusing on long-duration, score-based optimization problems, which have been underexplored compared to traditional short-duration contests.

**2. Real-World Relevance**: By using problems from the AtCoder platform, the benchmark reflects real-world optimization challenges faced in various industrial domains, such as logistics and scheduling.

**3. Comprehensive Evaluation**: The paper provides detailed metrics for assessing AI performance, including fine-grained and aggregated metrics that facilitate fair comparisons between AI systems and human participants.

**4. Interactive Framework**: ALE-Bench supports interactive agent architectures, enabling AI systems to leverage feedback and visualizations during the problem-solving process, akin to human strategies.

---

> ### Author Rebuttal · Authors · 2025-07-30
>
> We would like to express our deep gratitude for the valuable review of our research.
>
> First, we appreciate the reviewer's clear articulation of ALE-Bench's core strengths, and in particular:
>
> > ALE-Bench fills a gap in the current landscape of AI benchmarks by focusing on long-duration, score-based optimization problems,
>
> Recognizing this primary goal of our work is deeply encouraging.
> We also appreciate that the reviewer highlighted the design of ALE-Bench, which is interactive and allows for strategies similar to those used by humans:
>
> > ALE-Bench supports interactive agent architectures, enabling AI systems to leverage feedback and visualizations during the problem-solving process, akin to human strategies.
>
> Below we would like to address each concern the reviewer raised as "Limitations Weaknesses."
> Our response includes:
>
> - Dataset size with expansion plans and current sufficiency explanations (**[W1]**).
> - Contamination risks and monitoring plans (**[W2]**).
> - Performance variability through added metadata analysis (**[W3]**).
> - Paper scope clarification as benchmark contribution, positioning the exploration of different AI approaches as future work enabled by ALE-Bench (**[W4]**).
>
> Although the rebuttal format precludes the submission of a revised manuscript, we have incorporated all the revisions based on the reviewer's suggestion.
>
> ---
>
> ### [W1] Limited Dataset Size
>
> We are also aware of the modest dataset size.
> We plan to continue adding newly held contests to the dataset in the future.
> We have already reached an agreement with the AHC organizer to incorporate newly released contests, and the ALE-Bench framework has been engineered for seamless expansion.
>
> In addition, as mentioned in Section 6, the results of each contest are relatively reliable, which contributes to the stability of the overall results despite the limited size of the dataset.
> This is because each problem requires a process of trial and error.
> The variance from these trials is aggregated and smoothed out, thus lowering the variance of the final contest results.
> This is consistent with the fact that the abilities of human competitors are reliably evaluated with this number of problems.
>
> ---
>
> ### [W2] Contamination Risks
>
> As the reviewer noted, although no serious issues due to contamination have been confirmed at present (Section 5.4), there is a risk that they may arise in the future.
> In conjunction with our plans for adding new contests in the future (related to W1), we intend to establish a system for continuous monitoring.
> Additionally, with AtCoder's explicit permission, we deployed an AI agent in an actual AHC and have reported the results (Appendix C.5.2).
> Going forward, we are also considering performance monitoring under real-time, contamination-free conditions by continuous participation in live AHC.
>
> ---
>
> ### [W3] Performance Variability
>
> We thank the reviewer for the feedback regarding the performance variability across different problem types.
> We now make two points discussed on this topic explicit:
>
> 1. Contest duration (Sec 5.4, L279-): we note a clear performance difference between short-format contests (around 4 hours) and long-format contests (around 2 weeks), with higher performance in short-format contests.
> 2. Specific solution (Appendix C.3, L942-): we found that the problems where high performance was achieved, especially in the Iterative-Refinement setting, were mostly those where simulated annealing was effective.
>
> In addition to the above, we have now added metadata such as the problem characteristics and the algorithms used in top-level solutions, to enable a more detailed analysis.
>
> | Problem ID | Format | Judge Type | Top-Level Solution | Problem Genre |
> |:---|:---|:---|:---|:---|
> | ahc001 | Long | Standard | SA | Packing |
> | ahc002 | Short | Standard | SA | Routing |
> | ahc003 | Long | Reactive | Bayes | Inference |
> | ahc004 | Short | Standard | Adhoc | Packing |
> | ahc005 | Short | Standard | SA | Routing |
> | future-contest-2022-qual | Long | Reactive | Bayes, Evaluation | Scheduling |
> | ahc006 | Short | Standard | SA | Routing |
> | ahc007 | Short | Reactive | Adhoc | Network |
> | ahc008 | Long | Reactive | Adhoc, Structure | Game |
> | ahc009 | Short | Standard | SA, DP | Routing |
> | ahc010 | Short | Standard | SA | Network |
> | ahc011 | Long | Standard | Beam,SA | Puzzle |
> | ahc012 | Short | Standard | SA, Structure | Partitioning |
> | ahc014 | Long | Standard | SA | Puzzle |
> | ahc015 | Short | Reactive | Playout, Evaluation | Game |
> | ahc016 | Long | Reactive | Bayes, Structure | Inference |
> | ahc017 | Long | Standard | SA | Network |
> | ahc019 | Long | Standard | Beam, SA | Puzzle |
> | ahc020 | Short | Standard | SA | Covering |
> | ahc021 | Short | Standard | Beam | Planning |
> | toyota2023summer-final | Short | Reactive | Playout, Evaluation | Game |
> | ahc024 | Short | Standard | SA | Network |
> | ahc025 | Long | Reactive | Adhoc, Bayes | Partitioning |
> | ahc026 | Short | Standard | Beam, Evaluation | Planning |
> | ahc027 | Long | Standard | SA, Structure | Routing |
> | ahc028 | Short | Standard | DP, SA | Routing |
> | ahc030 | Long | Reactive | Bayes, SA | Inference |
> | ahc031 | Long | Standard | SA, Structure | Partitioning |
> | ahc032 | Short | Standard | Beam | Planning |
> | ahc033 | Long | Standard | Beam | Planning |
> | ahc034 | Short | Standard | Flow | Planning |
> | ahc035 | Short | Reactive | Evaluation | Game |
> | ahc038 | Long | Standard | Beam, Structure | Planning |
> | ahc039 | Short | Standard | SA | Partitioning |
> | ahc040 | Long | Reactive | Bayes, Beam | Packing |
> | ahc041 | Short | Standard | SA | Partitioning |
> | ahc042 | Short | Standard | Beam | Planning |
> | ahc044 | Short | Standard | SA | Scheduling |
> | ahc045 | Long | Reactive | Bayes, SA | Network |
> | ahc046 | Short | Standard | SA, Beam | Planning |
>
> *Solution algorithms*
>
> - SA (Simulated Annealing including hill climbing)
> - Beam (Beam Search including greedy approaches)
> - Bayes (Bayesian inference)
> - Adhoc (Ad-hoc approaches)
> - DP (Dynamic Programming)
> - Structure (Fixed structure assumption for large solution spaces)
> - Playout (Playout using random/greedy strategies)
> - Flow (Network flow)
> - Evaluation (Crafting good evaluation functions)
>
> *Problem genres*
>
> - Covering (finding coverage methods)
> - Routing (finding paths)
> - Partitioning (finding partition methods)
> - Game (dynamic game problems)
> - Puzzle (static game problems)
> - Scheduling (determining order)
> - Planning (finding action sequences)
> - Packing (finding packing methods)
> - Network (finding connectivity)
> - Inference (inference problems)
>
> From these analyses, we found that the AI systems achieved relatively high performance on routing-type problems, where the objective is to find optimal paths.
> These problems are well-studied in the literature, and the systems were able to apply standard neighborhood operations such as 2-opt effectively, which likely contributed to their success.
> In contrast, performance on planning-type problems, which involve generating sequences of actions, was lower.
> This appears to be due to the need for problem-specific neighborhood structures and evaluation functions, which are more difficult for the AI systems to construct automatically.
> This detailed analysis highlights the current capabilities and challenges of AI and further enhances the utility of our benchmark.
>
> We have added this metadata and discussion to the Appendix.
> We hope that these refined analyses serve as our response to the reviewer's concerns, making this research even more valuable.
>
> ---
>
> ### [W4] AI Approach Comparison
>
> Regarding the reviewer's point about the limited comparison of AI approaches, we recognize this as an important future research direction.
> At the same time, we see this as a good opportunity to clarify the scope of our research.
> This paper was submitted to the Dataset & Benchmark track, with our constructed ALE-Bench as the main contribution, and our primary focus is on demonstrating its utility.
> Comparing and exploring different AI approaches to achieve higher performance is a new research area that this benchmark opens up.
> We position our experiments as providing an "initial glimpse" into that field.
> With the reviewer's feedback in mind, we have clarified the positioning of this paper in the main text.
>
> Furthermore, in our experiments, we have evaluated three AI systems, particularly in the scaffolding setting: simple Iterative-Refinement, OpenHands, and the newly proposed ALE-Agent.
> Our primary objective was to demonstrate ALE-Bench's capability to evaluate diverse AI systems across a wide spectrum of complexity such as bare LLMs and complex AI agents.
> We have compared many of the current state-of-the-art models available, and by evaluating various AI systems from the One-Shot setting to those using scaffolding, we believe we have achieved the contribution within the scope of our paper.
>
> ---
>
> Once again, we would like to express our gratitude for the reviewer's contribution to making this research more meaningful and enhancing its clarity.

---

> > ### Comment · Reviewer_1rad · 2025-08-05
> >
> > I thank the authors for their detailed and thoughtful response. The rebuttal has significantly improved the clarity and presentation of their work, providing a clearer understanding of the main contributions and technical details. In addition, the authors have engaged in meaningful discussion regarding potential future research directions, which further strengthens the value of the paper.
> >
> > My concerns have been fully addressed, and I am pleased to increase my score accordingly.

---

### Official Review · Reviewer_ep2u · 2025-07-02

**Rating:** 4
**Confidence:** 3

**Summary:**

This paper proposed a new benchmark named ALE-Bench for long-horizon objective-driven algorithm engineering. It encompasses 49 tasks from online contests, which are NP-hard problems (probably without optimal solutions), and the LLM is requested to write code to solve these problems. Two modes are considered: a one-shot setting where an LLM can generate code only once, and a iterative-refinement setting where an LLM can make full use of a period of time (say 4 hours) to generate multiple codes and return the best one. Results show that modern LLMs can generate codes match the level of human novices to intermediates, and still have significant gaps to the level of human experts.

**Dataset Code Accessibility:**

Yes

**Ethical Considerations:**

No, there are no or only very minor ethics concerns

**Final Justification:**

In my initial review, I raised several concerns regarding related literature and experiment design. Authors provided detailed responses and new experimental results, which successfully addressed my concerns. Now I believe this is a valuable work to the evaluation of LLM on long range reasoning tasks.

**Limitations Weaknesses:**

1. Authors mentioned FunSearch as the only work on using LLM to generate code for solving NP-hard problems. However, this is not true. LLM for combinatorial optimization code generation is now an active topic (the FunSearch paper [Romera-Paredes2024] has been cited for more than 500 times), and quite a few works in this direction has been published, e.g., [Liu2024] and [Ye2024].

[Romera-Paredes2024] Romera-Paredes, B., Barekatain, M., Novikov, A., Balog, M., Kumar, M. P., Dupont, E., ... & Fawzi, A. (2024). Mathematical discoveries from program search with large language models. Nature, 625(7995), 468-475.

[Liu2024] Liu, F., Xialiang, T., Yuan, M., Lin, X., Luo, F., Wang, Z., ... & Zhang, Q. (2024, July). Evolution of Heuristics: Towards Efficient Automatic Algorithm Design Using Large Language Model. In International Conference on Machine Learning (pp. 32201-32223). PMLR.

[Ye2024] Ye, H., Wang, J., Cao, Z., Berto, F., Hua, C., Kim, H., ... & Song, G. (2024). ReEvo: Large Language Models as Hyper-Heuristics with Reflective Evolution. In The Thirty-eighth Annual Conference on Neural Information Processing Systems.

2. I do not quite understand Section 4. What is the purpose of developing this agent, and why it is helpful in terms of evaluating the code generation and reasoning ability of LLMs?

3. The designed evaluation protocol is somewhat questionable. First, using average performance as the main indicator, as mentioned by authors, is not enough. Conducting statistical significance test is a standard evaluation step in combinatorial optimization. Second, the comparison between LLM and human seems unfair. As also mentioned by authors, LLMs are much faster than human in generating codes, giving them unfair advantage if time budget is the only controlling variable. Moreover, considering that different LLMs have different inference efficiency (could be due to hardware differences), giving them the same time budget is also questionable. Maybe it is better to control the number of LLM calls.

4. The iterative-refinement setting is questionable. Different from the one-shot setting, it requires an improvement mechanism to let the LLM refine its code, which itself is a research topic [Zelikman2024]. How the dependence on a preset improvement mechanism affects the evaluation and conclusion?

[Zelikman2024] Zelikman, E., Lorch, E., Mackey, L., & Kalai, A. T. (2024). Self-taught optimizer (stop): Recursively self-improving code generation. In First Conference on Language Modeling.

4. The unique values of the proposed benchmark should be discussed deeper. What unique properties does it offer comparing to existing benchmarks? What novel conclusions can be obtained using this benchmark?

**Strengths Contributions:**

1. This paper contributes challenging tasks for evaluating the code generation capabilities of modern LLMs, which is of practical value considering that existing code benchmarks tend to become less challenging.

2. The large amount of human contestants in the platform offer high-quality data to evaluate the capabilities of modern LLMs with respect to human performance.

3. The conclusions are interesting and reveal limitations of existing LLMs in the code generation task.

---

> ### Author Rebuttal · Authors · 2025-07-30
>
> We sincerely thank the reviewer for the many constructive points raised.
>
> > ...which is of practical value considering that existing code benchmarks tend to become less challenging.
>
> Given the reviewer's understanding of the current situation of code benchmarks, we are grateful that our benchmark is evaluated as practical.
>
> > The conclusions are interesting and reveal limitations of existing LLMs in the code generation task.
>
> We are pleased that our conclusions were well-received by the reviewer.
>
> Regarding the "Limitations Weaknesses," we have:
>
> - Updated related work with suggested citations (**[W1]**).
> - Clarified the role of ALE-Agent as a strong baseline (**[W2]**).
> - Conducted statistical significance tests and analysis controlling for the number of LLM calls (**[W3]**).
> - Clarified the intent behind the Iterative-Refinement setting as a foundational evaluation (**[W4]**).
> - Expanded the discussion on the unique value of ALE-Bench (**[W5]**).
>
> Although the rebuttal format precludes the submission of a revised manuscript, we have incorporated necessary revisions based on the reviewer's suggestion.
>
> ---
>
> ### [W1] Related Works
>
> We thank the reviewer for pointing out these related works.
> We have carefully reviewed the suggested papers ([Liu2024], [Ye2024]) and found them very insightful, although they do not address problems from score-based contests.
> We would like to clarify that our claim in L90-91 was specifically about prior work on score-based coding contests, not the broader field of using LLMs for NP-hard problems.
> To better position our work, we have revised the paper to add a discussion of these valuable papers in Section 2 as follows:
>
> "While works like [Liu2024] and [Ye2024] also focus on solving NP-hard problems with LLMs, they do not address problems from score-based contests. Our benchmark is distinct in that it enables a large-scale performance comparison against thousands of human contestants, providing a unique perspective on the capabilities of LLMs."
>
> ---
>
> ### [W2] Purpose of ALE-Agent
>
> We thank the reviewer for suggesting a clearer explanation of the intention behind ALE-Agent.
> ALE-Agent serves as a specialized and powerful baseline for the research area opened by ALE-Bench.
> It incorporates established techniques like scaffolding, inference-time scaling, and domain knowledge, going beyond simple Self-Refine.
> Its significant performance improvement demonstrates ALE-Bench's ability to measure long-term problem-solving capabilities from bare LLMs to complex AI agents.
> This highlights that LLM abilities can be enhanced with such techniques, while also demonstrating ALE-Bench's effectiveness.
> We have clarified ALE-Agent's role in the main text.
>
> ---
>
> ### [W3] Evaluation Protocol
>
> We thank the reviewer for these two suggestions for making the experimental results more persuasive.
> We conducted additional experiments to address both points.
>
> **Statistical Significance**
>
> > Conducting statistical significance test is a standard evaluation step in combinatorial optimization.
>
> We agree and have performed five independent runs of the One-Shot setting (C++20) for several models, observing statistically significant differences.
>
> The table below shows the average performance with 95% confidence intervals (CIs).
> The Gemini 2.5 models are excluded as only their experimental models were available at the time of submission and discontinued on July 15, 2025.
>
> | Model | Average Performance |
> |:---|---:|
> | GPT-4.1 mini | 732.29 ± 27.90 |
> | GPT-4.1 | 780.81 ± 81.23 |
> | o3-high | 1057.81 ± 46.74 |
> | o4-mini-high | 882.13 ± 30.29 |
> | Gemini 2.0 Flash | 565.29 ± 55.97 |
> | Claude 3.7 Sonnet (Thinking) | 858.06 ± 17.95 |
> | DeepSeek-R1 | 854.65 ± 40.72 |
>
> We also conducted a Wilcoxon signed-rank test for model pairs.
> Each cell in the following table shows the p-value for a one-sided test where the null hypothesis (no difference) is tested against the alternative that the row model outperforms the column model.
> Values below 0.05 indicate statistically significant differences at the 95% confidence level.
> The results statistically suggest relationships (e.g., o3-high > o4-mini-high, etc.), validating that ALE-Bench can produce robust and differentiable measurements.
>
> |  | GPT-4.1 mini | GPT-4.1 | o3-high | o4-mini-high | Gemini 2.0 Flash | Claude 3.7 Sonnet (Thinking) | DeepSeek-R1 |
> |:---|---:|---:|---:|---:|---:|---:|---:|
> | GPT-4.1 mini | -- | 0.906 | 1.000 | 1.000 | 0.031 | 1.000 | 1.000 |
> | GPT-4.1 | 0.156 | -- | 1.000 | 1.000 | 0.031 | 1.000 | 1.000 |
> | o3-high | 0.031 | 0.031 | -- | 0.031 | 0.031 | 0.031 | 0.031 |
> | o4-mini-high | 0.031 | 0.031 | 1.000 | -- | 0.031 | 0.156 | 0.156 |
> | Gemini 2.0 Flash | 1.000 | 1.000 | 1.000 | 1.000 | -- | 1.000 | 1.000 |
> | Claude 3.7 Sonnet (Thinking) | 0.031 | 0.031 | 1.000 | 0.906 | 0.031 | -- | 0.594 |
> | DeepSeek-R1 | 0.031 | 0.031 | 1.000 | 0.906 | 0.031 | 0.500 | -- |
>
> **Controlling LLM Calls**
>
> > Maybe it is better to control the number of LLM calls.
>
> This is a valuable suggestion for a fairer comparison.
> We re-analyzed the Iterative-Refinement results by segmenting the average performance by the number of code generations.
> While our time-based evaluation mirrors the real-world contest environment (Appendix C.5.2), controlling for LLM calls offers a complementary perspective on pure model capability versus inference efficiency.
>
> The results are shown in the table below (rank percentiles in parentheses).
> The "Final" column shows the original setting's results (rerun, so values slightly changed compared to Table 3).
> A key finding is the progress difference between DeepSeek-R1 and GPT-4.1-mini.
> In early stages (fewer calls), there was a large performance gap of nearly 200 while their final results were similar.
> This is because DeepSeek-R1 managed to generate ~60 codes in four hours versus ~300 for GPT-4.1.
> This highlights the utility of evaluating by both wall-time and LLM calls (code generations).
>
> | Model | 10 | 30 | 50 | 100 | 200 | Final |
> |:---|---:|---:|---:|---:|---:|---:|
> | GPT-4.1 mini | 924 (81.08%) | 1010 (74.01%) | 1028 (71.94%) | 1109 (63.33%) | 1205 (53.06%) | 1218 (51.49%) |
> | o4-mini-high | 1262 (46.89%) | 1364 (35.59%) | 1419 (30.63%) | 1494 (24.37%) | 1522 (22.25%) | 1522 (22.25%) |
> | Gemini 2.5 Pro | 1212 (52.12%) | 1240 (49.05%) | 1314 (41.53%) | 1348 (37.48%) | 1351 (36.76%) | 1351 (36.76%) |
> | DeepSeek-R1 | 1107 (63.69%) | 1153 (58.06%) | 1215 (51.89%) | 1221 (51.13%) | 1221 (51.13%) | 1221 (51.13%) |
>
> These results have been added to the Appendix.
> We deeply appreciate the reviewer's valuable feedback to enhance the robustness and dimensions of our evaluation.
>
> ---
>
> ### [W4] Iterative-Refinement
>
> We agree that the mechanism for code improvement is a crucial research topic, from search methods to the information used for feedback and prompting, and it is currently being actively explored.
> However, as a benchmark paper, our goal is to show that ALE-Bench can evaluate long-term problem-solving capabilities promisingly and enables future research to determine state-of-the-art approaches ranging from bare LLMs to complex agents.
> Therefore, we focused on evaluating representative methods and our own ALE-Agent.
>
> We chose the Iterative-Refinement setting as it represents a fundamental and widely-adopted approach in the field.
> Its simplicity allowed us to run experiments on the full dataset (not just the lite version), enabling extensive analysis (contamination, plagiarism, and performance distribution comparisons with human participants) which led to its own subsection (Section 5.2), possibly causing confusion about its role.
>
> The intended structure of Chapter 5 was: *5.1 (One-Shot)* for basic capabilities, *5.2 (Iterative-Refinement)* for simple Self-Refine, and *5.3 (Scaffolding)* for advanced agents.
> We have added a sentence at the beginning of Section 5.2 to clarify this structural intent.
>
> ---
>
> ### [W5] Uniqueness of ALE-Bench
>
> We thank the reviewer for this valuable feedback, which will help us significantly improve the clarity of our paper.
> The primary unique value of ALE-Bench is that it measures an AI's ability in long-horizon (up to weeks), score-based algorithm engineering, a domain fundamentally different from existing benchmarks.
> While benchmarks like APPS, CodeContests, and LiveCodeBench have been pivotal, they evaluate performance on problems with known, exact solutions, grading submissions with a binary pass/fail metric.
> SWE-Bench and TheAgentCompany, although with longer horizons, also remain pass/fail evaluations.
> In contrast, ALE-Bench presents NP-hard optimization tasks with no single correct answer, encouraging iterative refinement and measuring continuous improvement.
> This enables novel insights into an AI's capacity for sustained reasoning and strategic planning, rather than merely one-shot code generation.
>
> While previous works like [Liu2024] and [Ye2024] also focus on solving NP-hard problems with LLMs, they do not address problems from score-based contests.
> Our benchmark is distinct in that it enables a large-scale performance comparison against thousands of human contestants, providing a unique perspective on the capabilities of LLMs.
> While MLE-Bench also adopts a score-based evaluation from competitions, it assesses data-centric machine learning skills in a GPU-intensive environment.
> In contrast, ALE-Bench is distinct in its focus on classical algorithm engineering for combinatorial optimization (e.g., routing, scheduling) in a resource-friendly CPU environment.
> We recognize that while these points are present in our Related Work section, we have revised the section to create a more precise map of the benchmark landscape, thereby better highlighting the unique research questions that ALE-Bench enables.
> Thank the reviewer again for helping us strengthen this key aspect of our contribution.
>
> ---
>
> We are confident that this review has deepened the substance of our research and thank the reviewer again for this opportunity.

---

> > ### Comment · Reviewer_ep2u · 2025-08-04
> >
> > I thank authors for the detailed respones. As my concerns have been addressed, I will increase my score.

---

### Official Review · Reviewer_rwsY · 2025-07-02

**Rating:** 4
**Confidence:** 3

**Summary:**

The paper introduces ALE-Bench, a new benchmark designed to evaluate AI systems on long-horizon, score-based algorithmic programming contests. Drawing from the AtCoder Heuristic Contests (AHC), the benchmark presents computationally hard optimization problems (e.g., routing, scheduling, planning) that lack known exact solutions. Unlike existing pass/fail coding benchmarks, ALE-Bench is designed for iterative solution refinement over extended periods.

**Dataset Code Accessibility:**

Yes

**Dataset Code Comments:**

The public release of the dataset, code, and environment setup scripts on platforms like GitHub and Hugging Face is commendable and promotes reproducibility.

**Ethical Considerations:**

No, there are no or only very minor ethics concerns

**Final Justification:**

I have read the author rebuttal and considered all raised points., I have engaged in discussions and responded to authors., I have filled in the "Final Justification" text box and updated "Rating" accordingly (before Aug 13) that will become visible to authors once decisions are released., I understand that Area Chairs will be able to flag up Insufficient Reviews during the Reviewer-AC Discussions and shortly after to catch any irresponsible, insufficient or problematic behavior. Area Chairs will be also able to flag up during Metareview grossly irresponsible reviewers (including but not limited to possibly LLM-generated reviews)., I understand my Review and my conduct are subject to Responsible Reviewing initiative, including the desk rejection of my co-authored papers for grossly irresponsible behaviors. https://blog.neurips.cc/2025/05/02/responsible-reviewing-initiative-for-neurips-2025/

**Limitations Weaknesses:**

The paper’s primary contribution is a dataset/framework; the agent techniques (prompting with domain knowledge, diversity-oriented best-first beam search) are incremental over Self-Refine and existing scaffolders.

The paper is motivated by the need to test advanced AI reasoning. However, the analysis suggests that the AI's success, particularly in shorter contests, may stem from a brute-force advantage rather than superior reasoning. This creates a tension regarding what the benchmark truly measures. If an AI can achieve a high score by testing 1,000 mediocre ideas in four hours, is it demonstrating "long-horizon problem-solving" or just a very efficient, massively parallel search? The benchmark's goal is to measure the former, but its structure may inadvertently reward the latter. This doesn't invalidate the benchmark, but it complicates the interpretation of its results as a pure measure of reasoning ability.

**Strengths Contributions:**

1. The paper addresses a well-recognized gap in AI evaluation. The community is moving beyond simple, short-horizon tasks towards more complex, end-to-end problems. ALE-Bench provides a robust, well-structured platform for driving research in this direction.
2. The benchmark is exceptionally well-designed. Its foundation on real-world AtCoder Heuristic Contests lends it credibility and ensures the problems are challenging and relevant.
3. The authors have conducted an extensive set of experiments on 22 different LLMs from major AI labs. The evaluation across different settings (one-shot vs. iterative), programming languages, and scaffolding methods (Self-Refine, OpenHands, ALE-Agent)  provides a comprehensive and valuable baseline for the community.

---

> ### Author Rebuttal · Authors · 2025-07-30
>
> We sincerely thank the reviewer for the deep understanding of our work and for the constructive feedback.
>
> > The paper addresses a well-recognized gap in AI evaluation. The community is moving beyond simple, short-horizon tasks towards more complex, end-to-end problems.
>
> We fully agree.
> Filling this gap has been our primary motivation, and we appreciate the reviewer's recognition of ALE-Bench's value in this regard.
>
> Additionally, as mentioned in the third point of "Strengths Contributions," we are grateful that the reviewer has also commended our comprehensive experimental study, which spans a diverse set of models and experimental setups.
>
> The points the reviewer listed as "Limitations Weaknesses" are also very well-founded and helpful.
> To address these concerns, we have:
>
> - Clarified that ALE-Agent was intentionally designed as a strong baseline to demonstrate the benchmark's capability (**[W1]**).
> - Conducted a new experiment showing ALE-Bench measures genuine long-horizon reasoning, not just brute-force parallel search (**[W2]**).
>
> Although the rebuttal format precludes the submission of a revised manuscript, we have incorporated all the revisions based on the reviewer's suggestion.
>
> ---
>
> ### [W1] Incremental Agent Techniques
>
> As the reviewer noted, ALE-Bench itself is the main contribution.
> ALE-Agent was introduced to initially explore the research opportunities opened up by ALE-Bench, as explained at the beginning of Section 4.
>
> Therefore, we strategically designed ALE-Agent as a strong and sophisticated baseline by incorporating established techniques like scaffolding, inference-time scaling, and domain knowledge.
> The substantial performance gains achieved by ALE-Agent underscore ALE-Bench's ability to effectively differentiate long-horizon problem-solving capabilities from basic LLMs to complex scaffolded AI agents.
> We are also confident that it will serve as an essential foundation for future, more advanced agent research.
>
> ---
>
> ### [W2] Brute-Force vs. Genuine Reasoning
>
> > If an AI can achieve a high score by testing 1,000 mediocre ideas in four hours, is it demonstrating "long-horizon problem-solving" or just a very efficient, massively parallel search? The benchmark's goal is to measure the former, but its structure may inadvertently reward the latter.
>
> We appreciate this insightful question, which reflects a profound understanding of the paper.
> We recognize this question is extremely important and central to clarifying what our benchmark truly measures.
> To investigate this, we conducted an additional experiment and obtained results suggesting that ALE-Bench has the potential to measure long-horizon problem-solving capabilities.
>
> **Experiment**
>
> To allow for a direct comparison between the Iterative-Refinement strategy and a strategy of parallel independent trials, thereby helping us understand what the benchmark measures, we conducted an experiment with the following setup.
> First, we selected six problems where o4-mini-high achieved the performance over 2000 in the Iterative-Refinement experiment.
> We assumed it would be difficult to discern the difference from a parallel search on problems where high performance was not already achieved.
> For these six problems, we ran the One-Shot (C++20) setting 150 times.
> This number of 150 was chosen to match the average number of code generations for o4-mini-high in the Iterative-Refinement setting.
>
> **Results**
>
> The experimental results are shown in the table below.
> The `Iterative-Refinement` column shows the performance value from the Iterative-Refinement setting (identical to Table A6), while the remaining columns list statistics from this additional experiment: the mean, sample standard deviation, minimum, maximum, and quartiles of the performance values (rounded to the nearest integer) for each problem.
> While a performance near 2000 was achieved on rare occasions, the vast majority of scores were much lower, and the difference between the maximum value and the Iterative-Refinement score was about 300 ($\approx 1 \sigma$) or more.
> This suggests that ALE-Bench demands true long-horizon reasoning capabilities to refine solutions based on feedback, rather than simply trying a large number of mediocre ideas.
>
> | Problem ID | Iterative-Refinement | Mean | Std. Dev. | Minimum | Q1 | Median | Q3 | Maximum |
> |:---|---:|---:|---:|---:|---:|---:|---:|---:|
> | ahc005 | 2107 | 1281 | 216 | 604 | 1138 | 1144 | 1492 | 1722 |
> | ahc006 | 2472 | 998 | 308 | 116 | 800 | 1017 | 1259 | 2174 |
> | ahc012 | 2236 | 445 | 329 | 123 | 123 | 466 | 697 | 1388 |
> | ahc020 | 2545 | 1061 | 104 | 579 | 1015 | 1015 | 1157 | 1731 |
> | ahc041 | 2306 | 988 | 251 | 444 | 852 | 1006 | 1074 | 1911 |
> | ahc044 | 2150 | 774 | 300 | -80 | 713 | 713 | 713 | 1831 |
>
> The reviewer's comment was very important and addressed a core issue.
> By conducting this additional experiment, we have not only directly addressed the reviewer's concern and underscored ALE-Bench's capacity to evaluate true long-horizon reasoning rather than mere brute-force exploration, but also further deepened the understanding and strengthened the case for our benchmark.
>
> ---
>
> Once again, we greatly appreciate the reviewer's thoughtful and incisive review.
> This feedback has been invaluable in strengthening both our paper and ALE-Bench.

---

### Decision · Program_Chairs · 2025-09-18

**Decision:**

Accept (poster)

**Comment:**

The paper introduces an LLM reasoning benchmark based on programming contests organized around the solution of NP-complete problems. The reviewers all agreed that the work represents a meaningful advance in the evaluation of long-term reasoning abilities of LLMs. The authors engaged deeply in the rebuttal period, both clarifying some discussion points and also strengthening their arguments via new experimental results. I recommend the paper's acceptance.

I have a few remaining concerns about the work. The authors might want to address them to the extent possible to maximize the impact of their research.

First, I question whether five independent runs are enough to assess statistical significance. (How can you estimate a meaningful 95% confidence interval with only five samples?) It's good to consider the issue of statistical significance, but simply running a Wilcoxon signed-rank test isn't enough.

Second, it seems to me the authors make too much of the distinction between binary pass/fail and "score-based" benchmarks. In a rebuttal they wrote "ALE-Bench presents NP-hard optimization tasks with no single correct answer": why isn't the optimal solution the correct answer? Why wouldn't binary pass/fail evaluation based on this optimal solution eventually teach an LLM well about these problems given a (probably unrealistically large) number of samples? Note that this isn't an argument that ALE-Bench isn't useful; it's an argument that it's not "unique".

Finally, I'm particularly concerned about the small size of the dataset. This suggests to me that ALE-Bench might quickly become contaminated as LLMs train on it, and indeed as they train on the publicly available data underlying the benchmark. We know from "no free lunch for optimization" style arguments that small datasets can be deeply unrepresentative of NP-hard optimization problems more broadly.